# Flexible neural representations of abstract structural knowledge in the human entorhinal cortex

Shirley Mark[1]*, Philipp Schwartenbeck[1,2], Avital Hahamy[2], Veronika Samborska[1], Alon Boaz Baram[2]*[†], Timothy E Behrens[1,2,3†]

[1]Wellcome Centre for Human Neuroimaging, University College London, London, United Kingdom; [2]Wellcome Centre for Integrative Neuroimaging, University of Oxford, Oxford, United Kingdom; [3]Sainsbury Wellcome Centre for Neural Circuits and Behaviour, University College London, London, United Kingdom

*For correspondence:
markshir@gmail.com (SM);
alon.baram@ndcn.ox.ac.uk (ABB)

[†]These authors contributed equally to this work

## eLife Assessment

Mark and colleagues developed and validated a **valuable** method for examining subspace generalization in fMRI data and applied it to understand whether the entorhinal cortex uses abstract representations that generalize across different environments with the same structure. The manuscript presents **convincing** evidence for the conclusion that abstract entorhinal representations of hexagonal associative structures generalize across different stimulus sets.

**Abstract** Humans' ability for generalization is outstanding. It is flexible enough to identify cases where knowledge from prior tasks is relevant, even when many features of the current task are different, such as the sensory stimuli or the size of the task state space. We have previously shown that in abstract tasks, humans can generalize knowledge in cases where the only cross-task shared feature is the statistical rules that govern the task's state–state relationships. Here, we hypothesized that this capacity is associated with generalizable representations in the entorhinal cortex (EC). This hypothesis was based on the EC's generalizable representations in spatial tasks and recent discoveries about its role in the representation of abstract tasks. We first develop an analysis method capable of testing for such representations in fMRI data, explain why other common methods would have failed for our task, and validate our method through a combination of electrophysiological data analysis, simulations, and fMRI sanity checks. We then show with fMRI that EC representations generalize across complex non-spatial tasks that share a hexagonal grid structural form but differ in their size and sensory stimuli, that is their only shared feature is the rules governing their statistical structure. There was no clear evidence for such generalization in EC for non-spatial tasks with clustered, as opposed to planar, structure.

## Introduction

If you grew up in a small town, arriving in a big city might come as a shock. However, you will still be able to make use of your previous experiences, despite the difference in the size of the environment: When trying to navigate the busy city streets, your knowledge of navigation in your hometown is crucial. For example, it is useful to know the constraints that a 2D topological structure exerted on distances between locations. When trying to make new friends, it is useful to remember how people in your hometown tended to cluster in groups, with popular individuals perhaps belonging to several groups. Indeed, the statistical rules (termed 'structural form', *Kemp and Tenenbaum, 2008*)

that govern the relationships between elements (states) in the environment are particularly useful for generalization to novel situations, as they do not depend on the size, shape, or sensory details of the environment (*Mark et al., 2020*). Such generalizable features of environments are proposed to be part of the 'cognitive map' encoding the relationships between their elements (*Tolman, 1948*; *Behrens et al., 2018*; *Mark et al., 2020*).

The most studied examples of such environments are spatial 2D tasks. In all spatial environments, regardless of their size or shape, the relations between states (in this case locations) are subject to the same Euclidean statistical constraints. The spatial example is particularly useful because neural spatial representations are well-characterized. Indeed, one of the most celebrated of these – grid cells in the entorhinal cortex (EC) – has been suggested as (part of) a neural substrate for spatial generalization (*Behrens et al., 2018*; *Whittington et al., 2022*). This is because (within a grid module) grid cells maintain their coactivation structure across different spatial environments (*Fyhn et al., 2007*; *Yoon et al., 2013*). In other words, the information embedded in grid cells generalizes across 2D spatial environments (including environments of different shapes and sizes). Following a surge of studies showing that EC spatial coding principles are also used in non-spatial domains (*Constantinescu et al., 2016*; *Garvert et al., 2017*; *Bao et al., 2019*; *Park et al., 2020*), we have recently shown that EC also generalizes over non-spatial environments that share the same statistical structure (*Baram et al., 2021*). Importantly, in that work, the graphs that described the same-structured environments were isomorphic – that is there was a one-to-one mapping between states across same-structure environments.

What do we mean when we say the EC has 'generalizable representations' in spatial tasks? and how can we probe these representations in complex non-spatial tasks? Between different spatial environments, each grid cell realigns: its firing fields might rotate and shift (*Fyhn et al., 2007*). Crucially, this realignment is synchronized within a grid module population (*Yoon et al., 2013*; *Gardner et al., 2022*), such that the change in the grid angle and phase of all cells is the same. This means that cells that have neighboring firing fields in one environment will also have neighboring firing fields in another environment – the coactivation structure is maintained (*Yoon et al., 2013*; *Gardner et al., 2022*). A mathematical corollary is that grid cells' activity lies in the same low-dimensional subspace (manifold, *Yoon et al., 2013*; *Gardner et al., 2022*) in all spatial environments. This subspace remains even during sleep, meaning the representation is stably encoded (*Burak and Fiete, 2009*; *Gardner et al., 2019*; *Trettel et al., 2019*).

We have recently developed an analysis method, referred to as 'subspace generalization', which allows for the quantification of the similarities between linear neural subspaces, and used it to probe generalization in cell data (*Samborska et al., 2022*). Unlike other representational methods for quantifying the similarity between activity patterns (like RSA, used in *Baram et al., 2021*; *Kriegeskorte et al., 2008*; *Diedrichsen and Kriegeskorte, 2017*), this method has the ability to isolate the shared features underlying tasks that do not necessarily have a straightforward cross-task mapping between states, such as when the sizes of tasks underlying graphs are different. Here, we use it to quantify generalization in such a case, but on fMRI data of humans solving complex abstract tasks rather than on cell data. We designed an abstract associative-learning task in which visual images were assigned to nodes on a graph and were presented sequentially, according to their relative ordering on the graph. The graphs belonged to two different families of graphs, each governed by a different set of statistical regularity rules (structural forms; *Kemp and Tenenbaum, 2008*) – hexagonal (triangular) lattice graphs, and community structure graphs. There were two graphs of each structural form. Crucially, the graph size and embedded images differed within a pair of graphs with the same structural form, allowing us to test generalization due to structural form across both environment size and sensory information.

We first validate our approach by showing that subspace generalization detects the known generalization properties of entorhinal grid cells and hippocampal place cells when rodents free-forage in two different spatial environments – properties that have inspired our study's hypothesis. Next, we propose that our method can capture these properties even in low-resolution data such as fMRI. We provide twofold support for this conjecture: through sampling and averaging of the rodent data to create low-resolution versions of the data, and through simulations of grid cells grouped into simulated voxels to account for the very low resolution of the BOLD signal. We use these simulations to discuss how the sensitivity of our method depends on various characteristics of the signal. Next, we

validate the method for real fMRI signals by showing it detects known properties of visual encoding in the visual cortex in our task. Finally, and most importantly, we show that EC generalizes its voxelwise covariance patterns over abstract, discrete hexagonal graphs of different size and stimuli, exactly as grid cells do in space. This result, however, did not hold for the community graph structures. We discuss some possible experimental shortcomings that might have led to this null result.

## Theory – 'subspace generalization'

How can we probe the neural correlates of generalization of abstract tasks in the human brain? Popular representational analysis methods such as RSA (*Kriegeskorte et al., 2008*; *Diedrichsen and Kriegeskorte, 2017*) and repetition suppression (*Grill-Spector et al., 2006*; *Barron et al., 2016*) have afforded some opportunities in this respect (*Baram et al., 2021*). However, because these methods rely on similarity measures between task states, they require labeling of a hypothesized similarity between each pair of states across tasks. Such labeling is not possible when we do not know which states in one task align with which states in another task. In the spatial example where states are locations, the mapping of each location in room A to locations in room B does not necessarily exist – particularly when the rooms differ in size or shape. This makes labeling of hypothesized similarity between each pair of locations impossible. How can we look for shared activity patterns in such a case?

We have recently proposed this can be achieved by studying the covariance of different neurons across states (*Samborska et al., 2022*) (as opposed to RSA – which relies on the correlation of different states across neurons). If two tasks contain similar patterns of neural activity (regardless of when these occurred in each task), then the *neuron × neuron* covariance matrix (across states within-task) will look similar in both tasks. This covariance matrix can be summarized by its principal components (PCs), which are patterns across neurons – akin to 'cell assemblies' – and their eigenvalues, which indicate how much each pattern contributes to the overall variance in the data. If representations generalize across tasks, then patterns that explain a lot of variance in task 1 will also explain a lot of variance in task 2. We can compute the task 2 variance explained by each of the PCs of task 1:

$$V_{12} = diag \left( PC_1^T A_2 A_2^T PC_1 \right)$$

where PC1 is a matrix with all task 1 PCs as its columns, ordered by their eigenvalues, and A2 is the *neurons × states* task 2 data. These PCs are ordered according to the variance explained in task 1. Hence, if the same PCs explain variance across tasks, early PCs will explain more variance in task 2 than late PCs. The cumulative sum of $V_{12}$ will be a concave function and the area under this concave function is a measure of how well neuronal patterns generalize across tasks (*Figure 1a*). We refer to this measure as subspace generalization.

As validation and demonstration of our method, we first use it to recover differences in generalization between grid cells and place cells in the rodent brain that have been shown previously with other methods. Next, we demonstrate the feasibility of our method in capturing this difference in generalization properties even after we manipulate the data and reduce its resolution. To complete the logical bridge from cells to voxels, we address the limitation of this demonstration: the low number of cells recorded. We simulate voxels from synthetic grid cells and show how our method's power depends on various characteristics of the signal. These analyses show that theoretically (and under reasonable conditions) our method could still detect medial temporal lobe generalization properties in fMRI BOLD signal. Finally, and most importantly, we use our method to analyze fMRI data, testing for generalization of the covariance between voxel representations in human EC across complex non-spatial graphs with common regularities – analogous to the generalization of grid cells in physical space. Crucially, in this task, other representational methods common in fMRI analysis, such as RSA or repetition suppression, would not be applicable (due to lack of one-to-one mapping between states across graphs), highlighting the usefulness of our method.

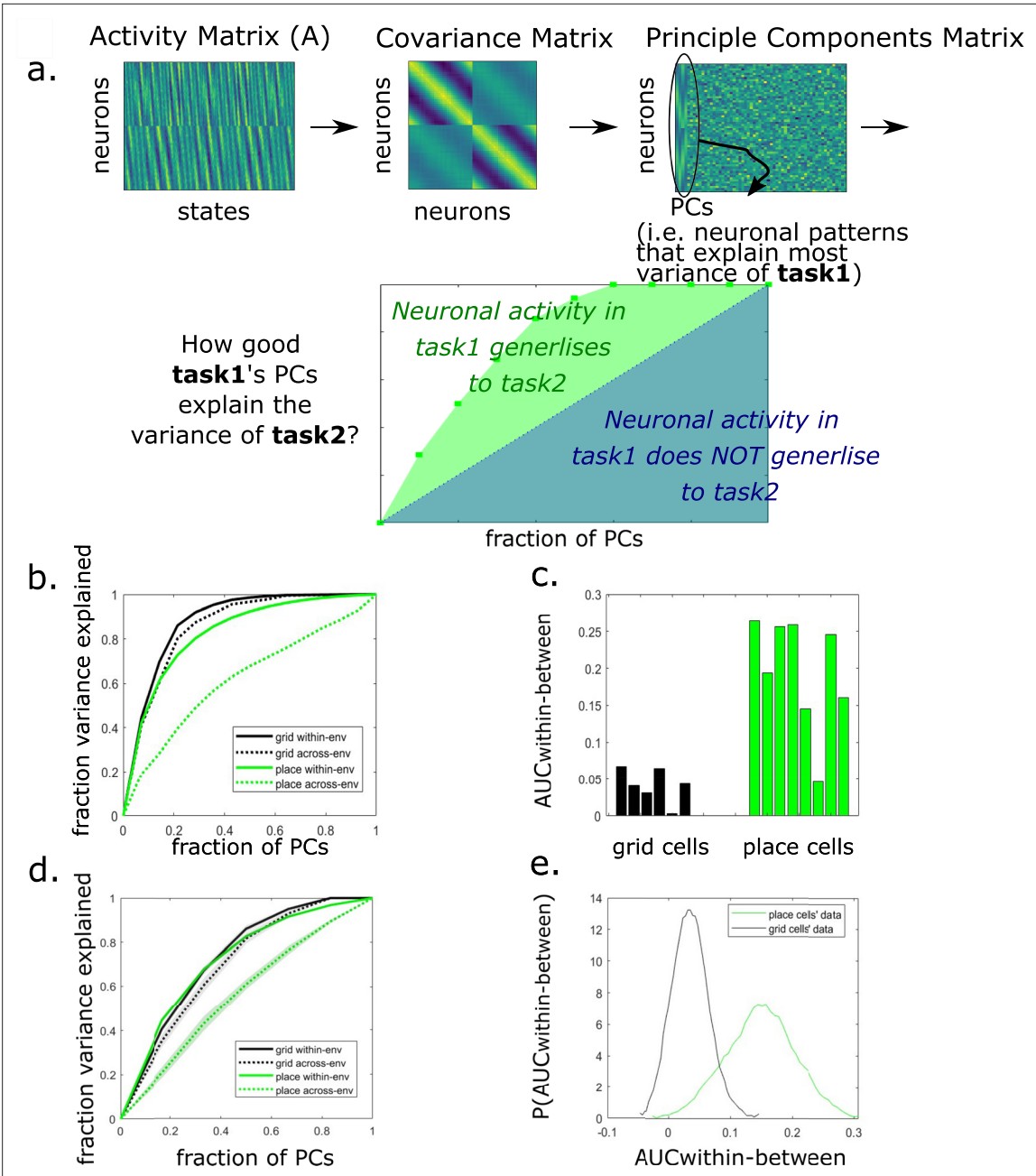

**Figure 1.** Subspace generalization across environments in grid and place cells in data from *Chen et al., 2018*. (**a**) Illustration of the subspace generalization method. The principal components (PCs) are calculated using the covariance matrix of the neuronal activity matrix. Then the activity matrix is projected on each PC (recorded when the animal was in the same or different environment/task) and the variance explained along each PC dimension is calculated. We calculate the area under the curve (AUC) of the cumulative sum of the variance explained on each PC's dimension as our similarity measure. When the similarity in neuronal patterns during the two different tasks is higher, the AUC is higher (green AUC is added to the blue AUC). (**b**) The cumulative variance explained by the PCs calculated using the activity of grid (black) or place (green) cells, within (solid lines) and across (dotted lines) environments. Subspace generalization is calculated as the difference between the AUC of two lines. The difference between the black lines is small, indicating generalization of grid cells across environments. The difference between the green lines is larger, indicating remapping of place cells (p < 0.001, permutation test, see Methods). (**c**) The difference between the within and across (solid and dashed lines in (a), respectively) environments AUCs of the cumulative variance explained by grid or place cells (black or green lines in (a), respectively). Data shown for all mice with enough grid or place cells (>10 recorded cells of the same type, each bar is a mouse and a specific projection (i.e. projecting on environment one or two)). The differences between the grid cells AUCs are significantly smaller than the place cells (p < 0.001 permutation test, see Appendix 1 for more statistical analyses and specific examples). (**d**) An example of the cumulative variance explained by the PCs, calculated using the constructed low-resolution version of grid and place cells data. The solid and dotted lines are average over 10 samples and the shaded areas represent the standard

*Figure 1 continued on next page*

*Figure 1 continued*

error of the mean across samples. Here, as above, the solid lines are projection within environment and the dotted lines are projections between environments. (**e**) Subspace generalization in the low-resolution version of the data captures the same generalization properties of grid vs place cells. The distributions were created via bootstrapping over cells from the same animal, averaging their activity, concatenating the samples across all animals, and calculating the AUC difference between within and across environments projections (p < 0.001 Kolmogorov–Smirnov test).

## Results

### Subspace generalization captures known generalization properties of grid and place cells

Grid cells and place cells differ in their generalization property. When an animal moves from one environment to another, place cells 'remap': they change their correlation structure such that place cells that are neighbors in environment 1 need not be neighbors in environment 2. By contrast, grid cells do not remap: the correlation structure between grid cells is preserved across environments, such that pairs of grid cells (within the same module) that have neighboring fields in environment 1 will also have neighboring fields in environment 2 (*Fyhn et al., 2007*). This is true even though each grid cell shifts and rotates its firing fields across environments – the grid cell population within a module realigns in unison (*Gardner et al., 2022*; *Waaga et al., 2022*). Crucially, the angle and phase of this realignment cannot be predicted in advance, meaning it is not possible to create hypotheses to test regarding the similarity between representations at a given location in environment 1 and a given location in environment 2 – a requirement for fMRI-compatible methods such as RSA or repetition suppression. In this section, we demonstrate how subspace generalization – which can also be useful in fMRI – captures the generalization properties of grid and place cells that have previously been shown only with traditional analysis methods that require access to firing maps of single cells.

We computed subspace generalization for grid and place cells recorded with electrophysiology in a previous study (*Chen et al., 2018*), in which mice freely foraged in two square environments: a real physical and a virtual reality (VR) (see Methods for more details). For our purposes, this dataset is useful because large numbers of both place cells and grid cells were recorded (concurrently within a cell type) in two different environments – rather than because of the use of a VR environment.

We compared two different situations: one where 'task 1' and 'task 2' were actually from the same environment, *Figure 1a* – solid line, within-environment and one where 'task 1' and 'task 2' were from different environments (*Figure 1a* – dotted line, across-environments).

As predicted, across environments, grid cells' subspaces generalized: PCs that were calculated using activity in one environment explained the activity variance in the other environment just as well as the within-environment baseline (*Figure 1a*, compare dotted and solid black lines, plots show the average of the projections of activity from one environment on EVs from the other environment and vice versa). The difference between the area under the curve (AUC) of the two lines was significantly smaller than chance (p < 0.001 using a permutation test, see Methods and *Appendix 1—figure 1*). Importantly, grid cells generalized much better between the environments than place cells; the difference in AUCs between the solid and dotted lines is significantly smaller for grid cells compared to place cells (*Figure 1b*, p < 0.001, for both permutation test and two-sample *t*-test, see Methods and Appendix 1). Interestingly, the difference in AUCs was also significantly smaller than chance for place cells (*Figure 1a*, compare dotted and solid green lines, p < 0.05 using permutation tests, see statistics and further examples in *Appendix 1—figure 2*), consistent with recent models predicting hippocampal remapping that is not fully random (*Whittington et al., 2020*).

### From neurons to voxels

So far, we have validated our method when applied to neurons. However, our primary interest in this manuscript is to apply it to fMRI data. To illustrate the efficacy of this approach in revealing generalizable neuronal subspaces within low-resolution data like fMRI, we applied our method to such data – both from manipulated electrophysiology and simulations. We first examined our method on low-resolution versions of the Chen et al. rodent MTL data, obtained by grouping and averaging cells. We show that our method can still detect subspace generalization even on the supra-cellular level. However, due to the small number of recorded cells, this analysis does not fully replicate a voxel's BOLD signal, which corresponds to the average activity of thousands of cells. To address this, we

simulated many grid cells and grouped them into voxels, with each voxel's activity corresponding to the average activity of its cells. We then applied subspace generalization to the simulated pseudo-voxels and examined how the results depend on various signal characteristics.

Using Chen et al. electrophysiology dataset, we first normalized each cell's firing rate maps, and then created bootstrapped low-resolution data: for each sampling iteration we sampled seven cells (with repeats) into two groups within each animal and averaged the activities of cells within each group. This results in a two-long vector for each animal. We then concatenate these vectors across animals. Note that for grid cells, this pooling over independent groups of neurons is reminiscent of pooling over different grid modules in a single subject. For each sample, we calculated the difference in the AUC between within and across environments projections as above (averaged over the projections on both environments, *Figure 1c*). We repeat this bootstrapping step to create a distribution of the differences in AUC for place cells and grid cells (*Figure 1d*). The difference in AUC was smaller for grid cells than for place cells (p < 0.001 Kolmogorov–Smirnov test), as is expected from the single cells' analysis above.

The required number of cells to simulate a voxel's activity (let alone multiple voxels) far exceeds the number of cells in the Chen et al. dataset. To overcome this limitation and support our conjecture that our method can detect subspace generalization even in fMRI BOLD signal, we next used simulated data. We simulated grid cells (see methods) organized into four grid modules, each composed of more than 10,000 cells. We organized the cells in each module into four groups (pseudo-voxels) and averaged the activity within each group (see Appendix 1 for an example of our analysis using different number of groups within each module, and how our results are affected by the number of voxels per module, *Appendix 1—figure 3*). We concatenated the pseudo-voxels from all modules into one vector and calculated the difference in subspace-generalization measure (i.e. the AUC of within and between environments). We explored how two characteristics of the data affect subspace generalization: whether the grouping into voxels (within each module) was organized according to grid phase, and the level of noise in the data.

We first grouped the cells into voxels randomly, that is without any a priori assumption on the relationship between the physical proximity of cells within the cortical layer and their firing rate maps. Examples of the resulted 'pseudo-voxels' activity maps can be seen in *Figure 2a*. However, recent work has suggested there is a relationship between grid cells' physical proximity and their grid phases (*Yi et al., 2018*). We therefore also simulated 'pseudo-voxels' by grouping grid cells, within each module, according to their grid phase (*Figure 2b*). The pseudo-voxel's signal in the latter case is substantially stronger (compare color bar scales a between *Figure 2a and b*).

How does the difference between the signal variances affect the subspace generalization measure? If the BOLD signal had no noise and all the cells within a voxel were indeed grid cells, the actual variance of the signal would not affect our measure (*Figure 2c*, the solid and dashed black lines are similar in both panels; that is, the PCs that explain the activity variance while the agent is in environment one explain the activity variance of environment two similarly well, no matter how the cells are sampled into voxels). However, this is, of course, unrealistic; the BOLD signal is noisy, and it is likely that voxel activity reflects non-grid cells activity as well. To address this, we incorporated noise into our simulated voxel's activity map. *Figure 2c* shows that increasing signal variance by grouping according to the grid phase leads to higher subspace generalization measure (AUC) compared to random sampling; random sampling results in small AUC ($AUC \approx 0.5$) which is close to the expected AUC following projections on random vectors (solid and dash blue lines in *Figure 2c*, left, see *Appendix 1—figure 3* for further analysis). Predictably, as the fraction of randomly sampled grid cells increases, the ability to detect subspace generalization in the presence of noise decreases (*Figure 2d*). Furthermore, sampling of grid cells according to phase increases the statistical power of the subspace generalization method when the amplitude of the noise increases (*Figure 2e*, *Appendix 1—figure 3*). To conclude, this shows under noisy conditions, if nearby grid cells have similar phase tuning, as has been shown (*Yi et al., 2018*), our method can in principle detect the generalization properties of grid cells, even in a very low-resolution data, akin to the fMRI BOLD signal. It can, in principle, work to detect generalization properties of any representation where nearby cells have similar tuning (such as orientation tuning in V1).

## Probing generalization across abstract tasks with shared statistical rules – task design and behavior

In human neuroimaging, the success of multivariate pattern analysis (*Haxby et al., 2001* and RSA, *Kriegeskorte et al., 2008*; *Diedrichsen and Kriegeskorte, 2017*) tells us that, as with cells, the

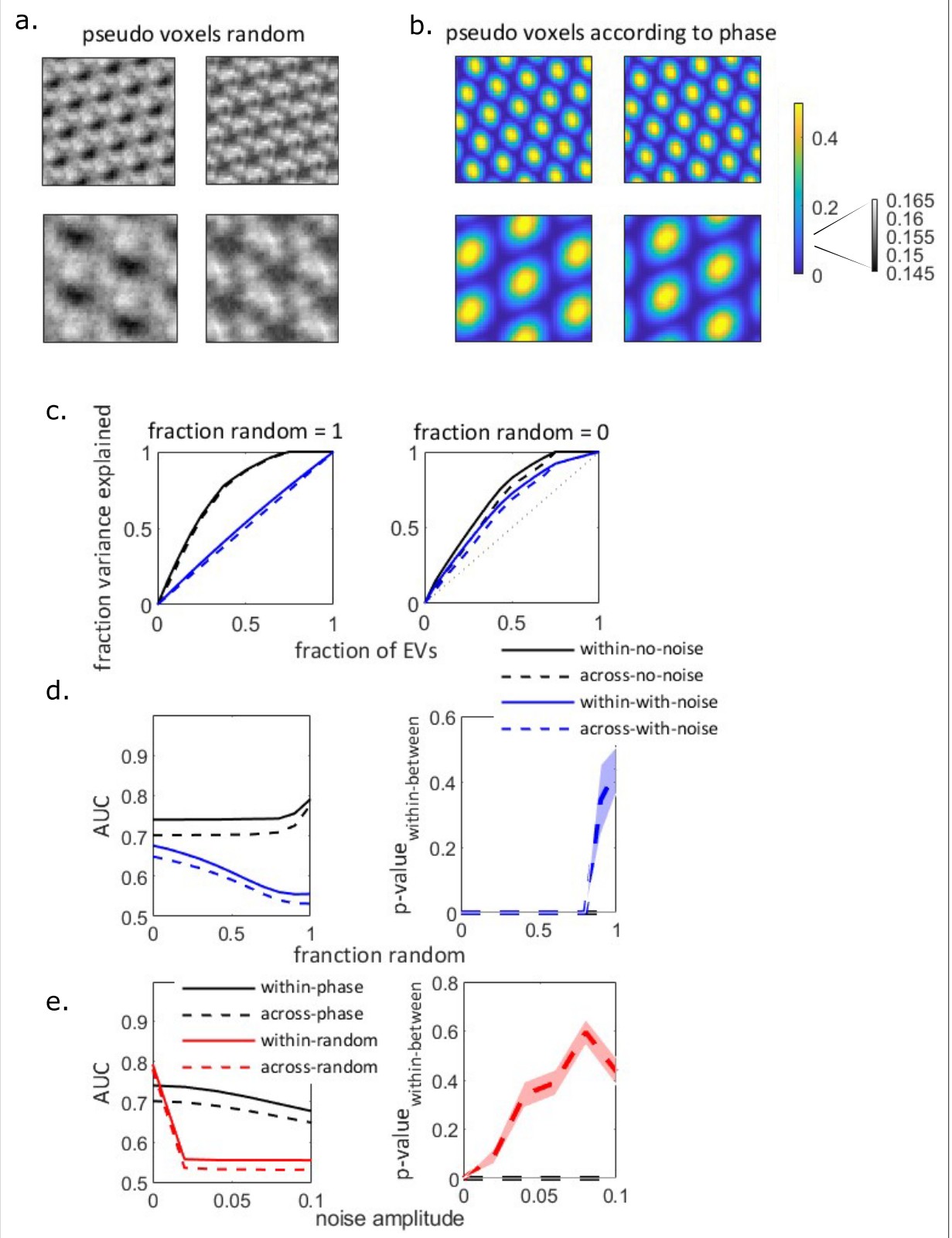

**Figure 2.** Simulated voxels from simulated grid modules. (**a**) Examples of simulated voxels activity map in the two environments, without noise. Upper: higher-frequency module, lower: lower-frequency module. Cells are grouped into voxels randomly. (**b**) Same as (**a**), but with cells grouped into voxels according to the grid phase. Note the different scale of the color bar between (**a**) and (**b**). (**c**) Subspace generalization plot for the 16 simulated voxels, where the grouping into voxels is either random (left) or according to phase (right). Legend as in (**d**), noise amplitude = 0.1. (**d**) Left: area under the

*Figure 2 continued on next page*

*Figure 2 continued*

curves (AUCs) of the subspace generalization plots in (c). as a function of the ratio of random vs phase-organized cells in the voxels, with no noise (black) or with high amplitude of noise (blue, noise amplitude = 0.1). Without noise (black lines), the subspace generalization measure (AUC) remains high even when the fraction of randomly sampled cells increases. However, in the presence of noise, the subspace generalization measure decreases with the fraction of randomly sampled cells. Right: p-value of the effect according to the permutation distribution (see methods, shaded area: standard error of the mean). In the presence of noise and when the cells are sampled randomly, AUCwithin-between becomes non-significant; see *Appendix 1—figure 3* for the dependency of the permutation distributions on the presence of noise and sampling. (**e**) Same as (d), except the continuous *X*-axis variable is the noise amplitude, for either of phase-organized (black) or randomly organized voxels (red). AUC decreases sharply with noise amplitude when the cells are sampled randomly, while it decreases more slowly when the cells are sampled according to phase. The decrease in AUC to chance level (i.e. AUC = 0.5) with the increase in noise amplitude results in insignificant difference in subspace generalization measure (AUCwithin-between). See *Appendix 1— figure 3* for the permutation distributions.

covariance between fMRI voxel activity contains information about the external world. It is therefore conceivable that we can measure the generalization of fMRI patterns across related tasks using the same measure of subspace generalization, but now applied to voxels rather than to cells. This will give us a measure of generalization in humans that can be used across tasks with no state-to-state mapping – for example when the size of the state space is different across tasks. In this section, we first describe the experimental paradigm we used to test whether, as in physical space, EC (1) generalizes over abstract tasks governed by the same statistical rules; and (2) does so in a manner that is flexible to the size of the environment. In the next section, we use known properties of visual encoding as a sanity check for the use of subspace generalization on fMRI data in this task. Finally, we describe how the fMRI subspace generalization results in EC depend on the statistical rules (structural forms) of tasks.

We designed an associative-learning task (*Figure 3A, B*, similar to the task in *Mark et al., 2020*) where participants learned pairwise associations between images. The images can be thought of as nodes in a graph (unseen by participants), where the existence of an edge between nodes translates to an association between their corresponding images (*Figure 3A*). There were two kinds of statistical regularities governing graph structures: a hexagonal/triangular structural form and a community structure. There were also two mutually exclusive image sets that could be used as nodes for a graph, meaning that each structural form had two different graphs with different image sets, resulting in a total of four graphs per participant. Importantly, two graphs of the same structural form were also of different sizes (36 and 42 nodes for the hexagonal structure; 35 and 42 nodes for the community structure – 5 or 6 communities of 7 nodes per community, respectively), meaning states could not be aligned even between graphs of the same structural form. The pairs of graphs with the (approximately) same sizes across structural forms used the same visual stimuli set (*Figure 3B*). This design allowed us to test for subspace generalization between tasks with the same underlying statistical regularities, controlling for the tasks' stimuli and size.

Participants were trained on the graphs for 4 days, and graph knowledge was assessed in each of the days using a battery of tests described previously (*Mark et al., 2020* and methods). Some tests probed knowledge of pairwise (neighboring) associations (*Figure 3C, D*) and others probed 'a sense of direction' in the graph, beyond the learned pairwise associations of neighboring nodes (*Figure 3E, F*). In all tests, the performance of participants improved with learning and was significantly better than chance by the end of training (*Figure 3C–F*), suggesting that participants were able to learn the graphs and developed a sense of direction even though they were never exposed to the graphs beyond pairwise neighbors. Note that while all participants performed well on tests of neighboring associations, the variance across participants for tests of non-neighboring nodes was high, with some participants performing almost perfectly and others close to chance (compare panels C, D to panels E, F). At the end of the training days, we asked participants whether they noticed how the images were associated with each other. Twenty-six out of 28 participants recognized that in two sets, the pictures were grouped.

## fMRI task and analysis

On the fifth day, participants performed a task in the fMRI scanner. Each block of the scan included one of the four graphs the participant has learned and started with a self-paced image-by-image random walk on the graph to allow inference of the currently relevant graph (*Figure 4a*, data not used in this manuscript). The second part of the block had two crucial differences. First, images were

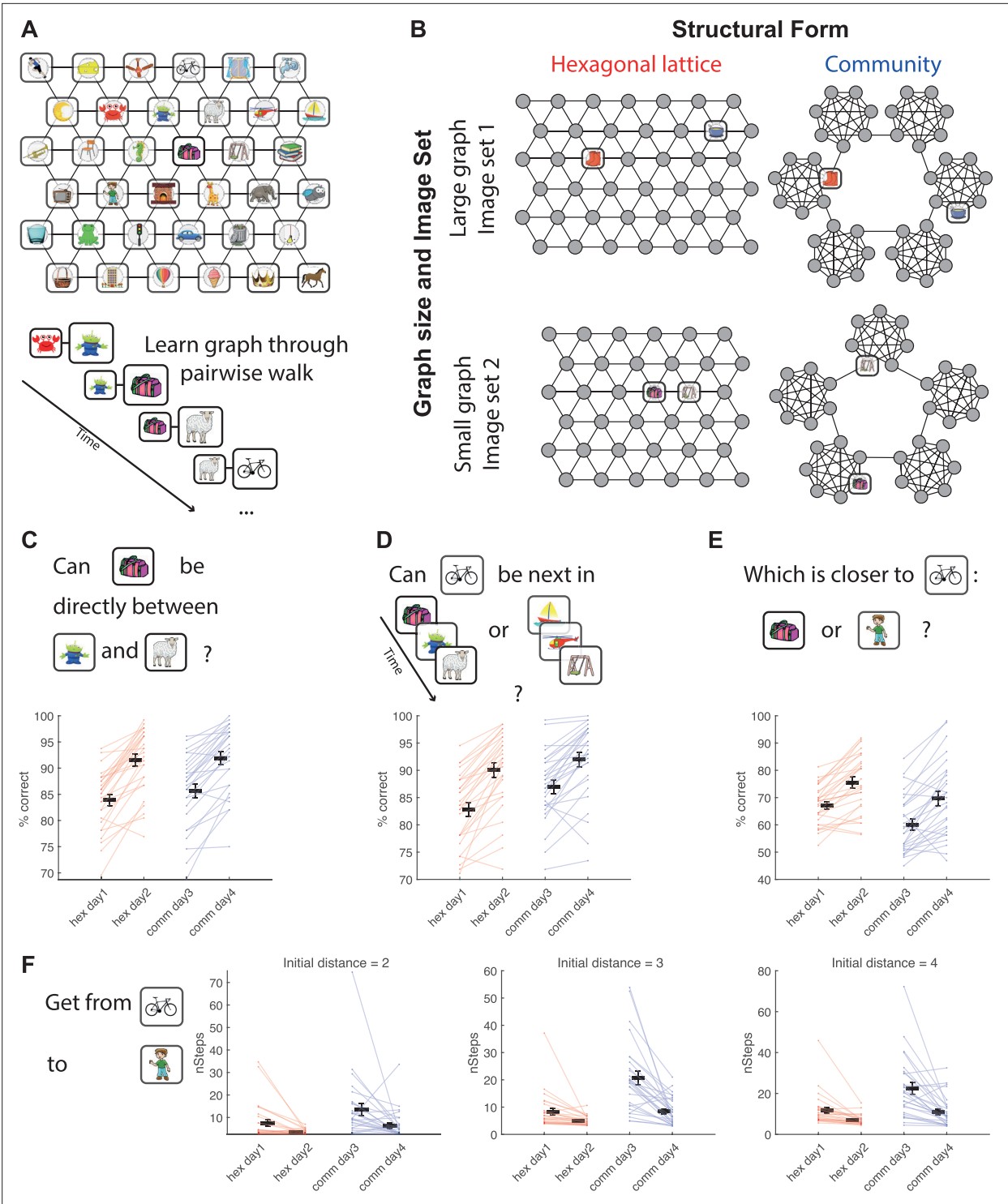

**Figure 3.** Experimental design and behavior. (**A**) Example of an associative graph. Participants were never exposed to this top-down view of the graph – they learned the graph by viewing a series of pairs of neighboring images, corresponding to a walk on the graph. To aid memorization, we asked participants to internally invent stories that connect the images. (**B**) Each participant learned four graphs: two with a hexagonal lattice structure (both learned on days 1 and 2) and two with a community structure (both learned on days 3 and 4). For each structural form, there was one larger graph and one smaller graph. The nodes of graphs with approximately the same size were drawn from the same set of images. (**C–F**) In each day of training, we used four tests to probe the knowledge of the graphs, as well as to promote further learning. In all tests, participants performed above chance level on all days and improved their performance between the first and second days of learning a graph. (**C**) Participants were asked whether an image X can appear between images Y and Z (one-sided *t*-test against chance level (50%): hex day 1 *t*(27) = 31.2, p < 10⁻²²; hex day 2 *t*(27) = 35.5, p < 10⁻²³; comm

*Figure 3 continued on next page*

*Figure 3 continued*

day 3 $t(27)$ = 26.9, p < $10^{-20}$; comm day 4 $t(27)$ = 34.2, p < $10^{-23}$; paired one-sided *t*-test between first and second day for each structural form: hex $t(27)$ = 4.78, p < $10^{-5}$; comm $t(27)$ = 3.49, p < $10^{-3}$). (**D**) Participants were shown two three-long image sequences and were asked whether a target image can be the fourth image in the first, second, or both of the sequences (one-sided *t*-test against chance level (33.33%): hex day 1 $t(27)$ = 39.9, p < $10^{-25}$; hex day 2 $t(27)$ = 42.3, p < $10^{-25}$; comm day 3 $t(27)$ = 44.8, p < $10^{-26}$; comm day 4 $t(27)$ = 44.2, p < $10^{-26}$; paired one-sided *t*-test between first and second day for each structural form: hex $t(27)$ = 3.97, p < $10^{-3}$; comm $t(27)$ = 2.81, p < $10^{-2}$). (**E**) Participants were asked whether an image X is closer to image Y or images Z, Y and Z are not neighbors of X on the graph (one-sided *t*-test against chance level (50%): hex day 1 $t(27)$ = 12.6, p < $10^{-12}$; hex day 2 $t(27)$ = 12.5, p < $10^{-12}$; comm day 3 $t(27)$ = 5.06, p < $10^{-4}$; comm day 4 $t(27)$ = 7.42, p < $10^{-7}$; paired one-sided *t*-test between first and second day for each structural form: hex $t(27)$ = 3.44, p < $10^{-3}$; comm $t(27)$ = 2.88, p < $10^{-2}$). (**F**) Participants were asked to navigate from a start image X to a target image Y. In each step, the participant had to choose between two (randomly selected) neighbors of the current image. The participant repeatedly made these choices until they arrived at the target image (paired one-sided *t*-test between number of steps taken to reach the target in first and second day for each structural form. Left: trials with initial distance of two edges between start and target images: hex $t(27)$ = 2.57, p < $10^{-2}$; comm $t(27)$ = 2.41, p < $10^{-2}$; middle: initial distance of three edges: hex $t(27)$ = 2.58, p < $10^{-2}$; comm $t(27)$ = 4.67, p < $10^{-2}$; right: trials with initial distance of four edges: hex $t(27)$ = 3.02, p < $10^{-2}$; comm $t(27)$ = 3.69, p < $10^{-3}$). Note that while feedback was given for the local tests in panels C and D, no feedback was given for the tests in panels E and F to ensure that participants were not directly exposed to any non-local relations. The location of different options on the screen was randomized for all tests. Hex: hexagonal lattice graphs. Comm: community structure graphs.

arranged into sequences of three images that were presented in rapid succession, corresponding to a walk of length 3 on the graph (***Figure 4b*** and ***Appendix 1—figure 5*** for the partitioning the graphs into three-image sequences). The time between two successive sequences was 800 ms (***Figure 4c***). Second, while the order within each three-image sequence was dictated by the graph, the order across the sequences was pseudo-random. We needed this second manipulation to ensure coverage of the graph in every block and to eliminate the possibility of spurious temporal correlations between neighboring sequences. However, if we had presented images individually in this random order, graphs with the same stimuli set would have been identical, making it difficult for subjects to maintain a representation of the current graph across the block. While the images were the same across 2 graphs, the sequences of neighboring images uniquely identified each graph, inducing a sensation of 'moving' through the graph. To encourage attention to the neighborhood of the sequence in the graph, in 12.5% of trials, the sequence was followed by a single image ('catch trial' in ***Figure 4c***), and participants had to indicate whether it was associated with the last image in the sequence (***Figure 4c***). Participants answered these questions significantly better than chance (***Appendix 1—figure 6***), indicating that they indeed recognize the correct graph and maintain the correct representation during the block (*t*-test, p < 0.001 for both structures, $t[27]_{hex}$ = 11.3, $t[27]_{comm}$ = 10.6). At the end of each block, participants were asked whether they recognized which images set they currently observed (see Method and Appendix 1 for more details). Participants answered these questions significantly better than chance (*t*-test, p < 0.001 for both structures, $t[27]_{hex}$ = 3.8, $t[27]_{comm}$ = 9.96, see ***Appendix 1—figure 6***), again indicating that they correctly recognized the current graph in the scanner.

To analyze this data, we used the subspace generalization method as described for the rodent data but replacing the firing of neurons at different spatial locations with the activity of fMRI voxels for different three-image sequences. To do this, we first performed a voxelwise GLM where each regressor modeled all appearances of a particular three-image sequence in a given run, together with several nuisance regressors (see Methods). This gave us the activity of each voxel for each sequence. For each voxel, in each run, we extracted the 100 nearest voxels and formed a matrix of sequence × voxels. These are analogous to the data matrices, **B,** in ***equation 1***. We then computed subspace generalization using the PCs of the voxel × voxel covariance matrix instead of the cell × cell covariance matrix (***Figure 4d***).

We then employed a leave-one-out cross-validation by repeatedly averaging the activation matrices from three runs of graph X, calculating the PCs from this average representation, and then projecting the activation matrix of the held-out run of control graph X (or a test graph Y) on these PCs. This ensures that the 'PC' and 'data' graphs are always from different runs. We then calculated the subspace generalization between each pair of graphs, resulting in a 4 × 4 matrix at each voxel of the brain (***Figure 4d***).

We refer to the elements of this 4 × 4 matrix in the following notation: we denote by H/C graphs of either hexagonal or community structure, and by s/l either small or large stimuli sets (matched across graphs of different structures). For example, HsCs denotes the element of the matrix corresponding

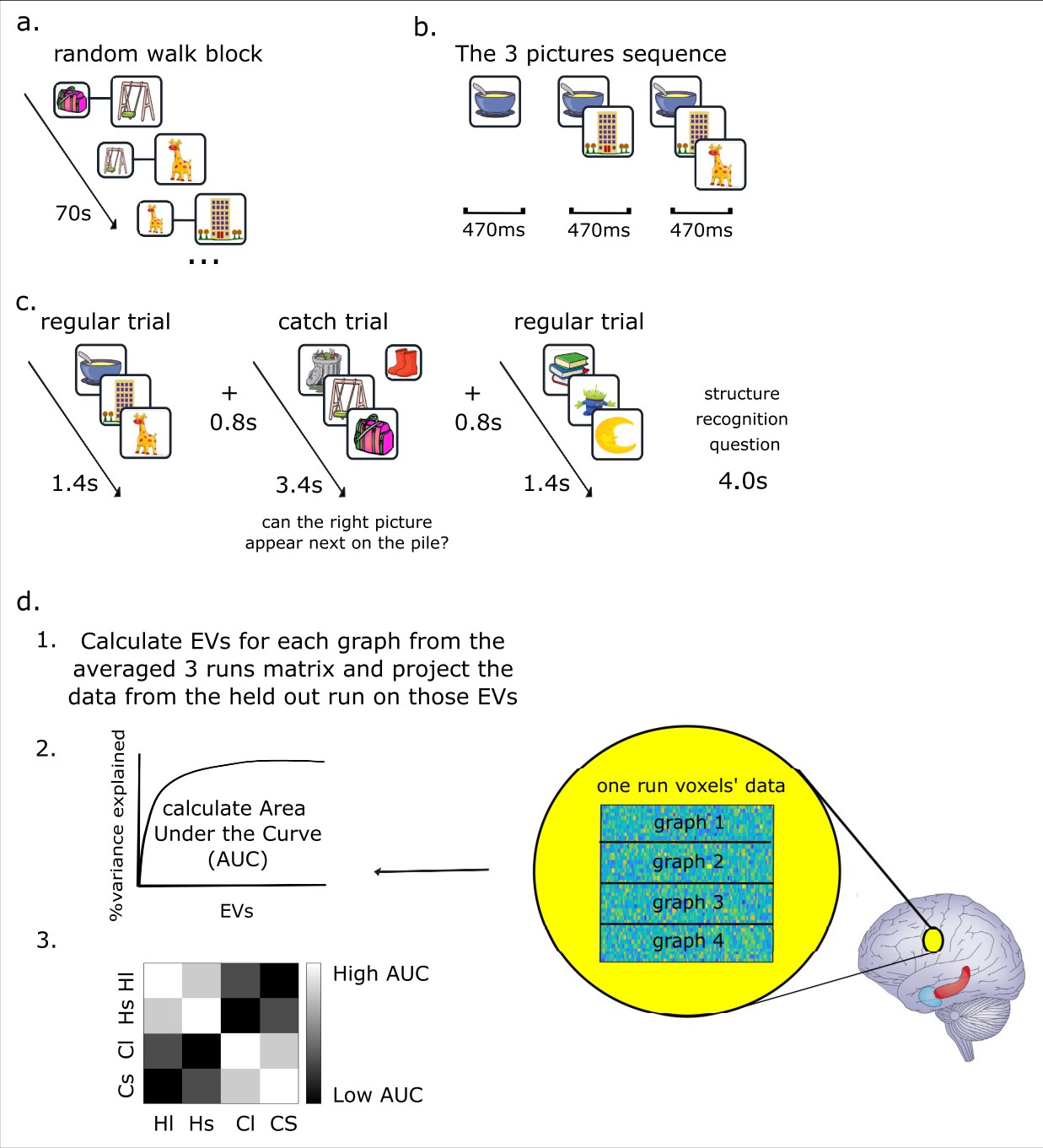

**Figure 4.** fMRI experiment and analysis method (subspace generalization). (**a**) Each fMRI block starts with 70 s of random walk on the graph: a pair of pictures appears on the screen, each time a participant presses enter a new picture appears on the screen and the previous picture appears behind (similar to the three pictures sequence, sell below). During this phase, participants are instructed to infer which 'pictures set' (i.e. graph) they are currently playing with. Note that fMRI data from this phase of the task is not included in the current manuscript. (**b**) The three pictures sequence: three pictures appear one after the other, while previous picture/s still appear on the screen. (**c**) Each block starts with the random walk (panel a). Following the random walk, sequences of three pictures appear on the screen. Every few sequences, there was a catch trial in which we asked participants to determine whether the questioned picture can appear next on the sequence. (**d**) Subspace generalization method on fMRI voxels. Each searchlight extracts a beta X voxels' coefficients (of three-image sequences) matrix for each graph in each run (therefore, there are four such matrices). Then, using cross-validation across runs, the left-out run matrix of one graph is projected on the EVs from the (average of three runs of the) other graph. Following the projections, we calculate the cumulative percentage of variance explained and the area under this curve for each pair of graphs. This leads to a 4 × 4 subspace generalization matrix that is then being averaged over the four runs (see main text and methods for more details). The colors of this matrix indicate our original hypothesis for the study: that in EC, graphs with the same structure would have larger (brighter) area under the curves (AUCs) than graphs with different structures (darker).

to activity from the small hexagonal graph projected on PCs calculated from the small (same image set) community-structure graph.

## Testing subspace generalization on visual representations

To verify our analysis approach is indeed valid when used on our fMRI data, we first tested it on the heavily studied object encoding representations in lateral occipital cortex (LOC, *Malach et al., 1995* PNAS, Grill-Spector). Recall that our stimuli in the scanner were concurrently presented sequences of three images of objects. We reasoned that these repeated sequences would induce correlations between object representations that should be observable in the fMRI data and detectable by our method. This would allow us to identify visual representations of the objects without ever specifying when the stimuli (i.e. three-image sequences) were presented.

To this end, we compared subspace generalization computed between different runs that included the same stimuli (three-image sequences, with different order across sequences between runs) with subspace generalization computed between runs of different stimuli while controlling for the graph structure. This led to the contrast [HlHl + ClCl + HsHs+ CsCs] − [HlHs + HsHl + ClCs + CsCl], which had a significant effect in LOC (*Figure 5a*), peak MNI [−44, −86, −8], $t(27)\_peak = 4.96$, p_tfce <0.05 based on an FWE-corrected nonparametric permutation test, corrected in bilateral LOC mask (Harvard-Oxford atlas, *Desikan et al., 2006*, Neuroimage). In an additional exploratory analysis, we tested the significance of the same contrast in a whole-brain searchlight. While this analysis did not reach significance once corrected for multiple comparisons, the strongest effect was found in LOC (*Figure 5a*). Note that in this contrast, we intentionally ignored the elements of the $4 \times 4$ matrix where the data and the PCs came from graphs with the same images set and a different structure (HlCl, HsCs, ClHl, and CsHs), because they did not share the exact same visual stimuli (the three-image sequence). In these cases, we did not have a hypothesis about the subspace generalization in LOC. These results suggest that we can detect the correlation structure induced by stimuli without specifying when each stimulus was presented.

## EC generalizes a low-dimensional representation across hexagonal graphs of different stimuli and sizes

Having established that the subspace generalization method can detect meaningful correlations between fMRI voxels, we next aimed to test whether EC will represent the statistical structure of abstract graphs with generalizable low-dimensional representations. We first tested this for discretized 2D (hexagonal) graphs, using the community structure graphs as controls: We tested whether the EC subspaces from hexagonal graph blocks were better aligned with the PCs of other hexagonal blocks than with the PCs from community graph blocks, i.e. ([HlHl + HlHs + HsHl + HsHs] − [HlCl + HlCs + HsCl + HsCs], *Figure 5b*). This contrast was significant in the right EC peak MNI [28, −10, −40], $t(27)\_peak = 4.2$, p_tfce <0.01 based on an FWE-corrected nonparametric permutation test, corrected in a bilateral EC mask (*Figure 5b*) (Jülich atlas, *Amunts et al., 2005*). We obtained a null result for the equivalent analysis for community structure graphs ([ClCl + ClCs + CsCl + CsCs] − [ClHl + ClHs + CsHl + CsHs]). This was particularly due to low subspace generalization across different runs of the same community structure graphs (bottom two diagonal elements in *Figure 5b* right, compare to our original hypothesis subspace generalization matrix in *Figure 4d*). See the Discussion for possible interpretations of this null result.

To ensure the robustness of the hexagonal graphs result, we next tested the same effect in an orthogonal ROI from our previous study. In *Baram et al., 2021*, we have shown that EC generalizes over different reinforcement learning tasks with the exact same structure. We therefore tested the same effect in that ROI (all voxels in the green cluster in Figure 3D in *Baram et al., 2021*, peak MNI: [25, −5, −28]), and indeed the [HlHl + HlHs + HsHl + HsHs] − [HlCl + HlCs + HsCl + HsCs] contrast was significant (one-sided *t*-test, $t(27) = 3.6$, p < 0.001, *Figure 5c*).

Taken together, these results suggest that as in physical space, different abstract hexagonal graphs are being represented on the same EC low-dimensional subspace. This is consistent with a view where the same EC cell assembly represents both hexagonal graphs, and that these cells covary together – even when the underlying size of the graph is different.

## Discussion

The contributions of this manuscript are twofold: first, we show that EC representations generalize over hexagonal abstract graphs of different sizes, highlighting the importance of the statistical

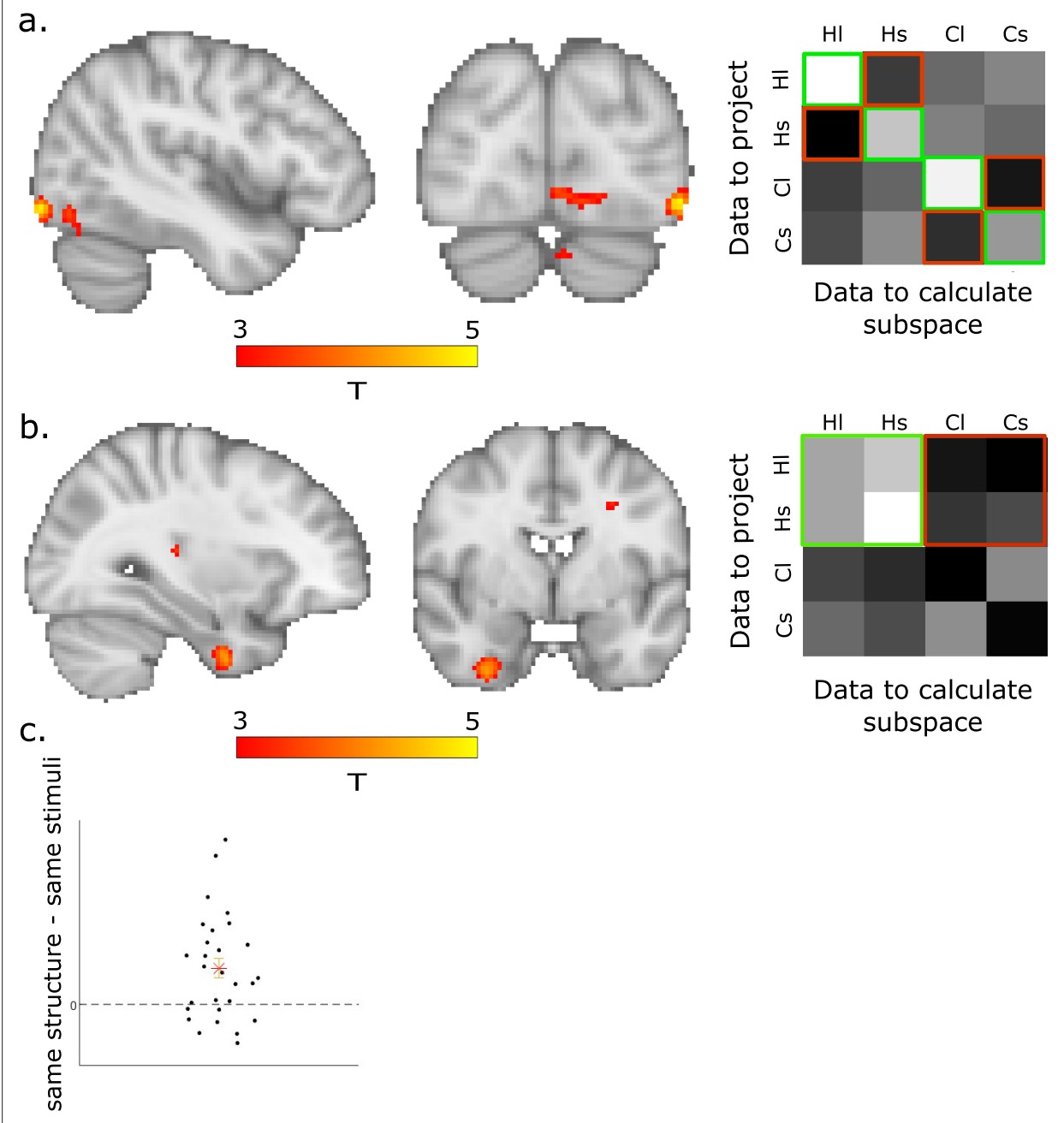

**Figure 5.** Subspace generalization in visual and structural representations. (**a**) Subspace generalization of visual representations in lateral occipital cortex (LOC). Left: difference in subspace generalization was computed between different blocks that included the same stimuli with subspace generalization computed between blocks of different stimuli while controlling for the graph structure, that is [HlHl +ClCl + HsHs + CsCs] − [HlHs + HsHl + ClCs + CsCl]. $t(27)\_peak$ = 4.96, p_tfce <0.05 over LOC. Right: visualization of the subspace generalization matrix averaged over all LOC voxels with $t > 2$ for the [HlHl +ClCl + HsHs + CsCs] − [HlHs + HsHl + ClCs + CsCl] contrast, that is green minus red entries. (**b**) Entorhinal cortex (EC) generalizes over the structure of hexagonal graphs. Left: the effect for the contrast [HlHl + HlHs + HsHl + HsHs] − [HlCl + HlCs + HsCl + HsCs], that is the difference between subspace generalization of hexagonal graphs data, when projected on principal components (PCs) calculated from (cross-validated) hexagonal graphs (green elements in right panel) vs community structure graphs (red elements). $t(27)\_peak$ = 4.2, p_tfce <0.01 over EC. Right: same as in (a) right but for the [HlHl + HlHs + HsHl + HsHs] − [HlCl + HlCs + HsCl + HsCs] contrast in EC. (**c**) The average effect in an ROI from Baram et al. (green cluster in Figure 3D of **Baram et al., 2021**) for each participant. Star denotes the mean, error bars are SEM.

properties of the environment to generalization. This expands our previous work (both experimental, *Baram et al., 2021* and theoretical, *Whittington et al., 2020*), suggesting EC plays an important role in generalization over abstract tasks, to the case where the tasks are governed by the same statistical rules but are not governed by the exact underlying graph (transition structure). This view builds on the known generalization properties of EC in physical space (*Fyhn et al., 2007*; *Gardner et al., 2022*) and on recent literature highlighting parallels between medial temporal lobe representations in spatial and non-spatial environments (*Behrens et al., 2018*; *Whittington et al., 2022*). Second, we present an fMRI analysis method (subspace generalization), adapted from related work in electrophysiology analysis (*Samborska et al., 2022*), to quantify generalization in cases where a mapping between states across environments is not available (though see *Yi et al., 2018* for our previous fMRI application of this method in the visual domain).

Exploiting previous knowledge while making decisions in new environments is a hard challenge that humans and animals face regularly. To enable generalization from loosely related previous experiences, knowledge should be represented in an abstract and flexible manner that does not depend on the particularities of the current task. Understanding the brain's solution to this computational problem requires a definition of a 'generalizable representation', and a way of quantifying it. Here, we define generalization as sharing of neuronal manifold across representations of related tasks. The particular assumption here is that in the EC, such manifolds encode the relevant information about the particular structural form of the task.

An example of such generalization has previously been observed in the spatial domain, in grid cells recordings across different physical environments, regardless of shape or size (*Fyhn et al., 2007*; *Gardner et al., 2022*). This was usually done through direct comparison of the pairwise activity patterns of cells (*Fyhn et al., 2007*; *Yoon et al., 2013*; *Gardner et al., 2022*). However, this is not possible to do in fMRI, rendering the examination of EC generalization in complex abstract tasks difficult. 'Subspace generalization' relies on the idea that similarity in activity patterns across tasks implies similarity of the within-task correlations between neurons. These are summarized in the similarity between the (low-dimensional) linear subspaces where the activity of the neurons/voxels representing the two tasks lies. For fMRI purposes, this similarity between within-task neuronal correlations should be reflected in the similarity between within-task correlations across voxels, as long as the relevant neurons anatomically reside across a large enough number of voxels. Importantly, comparing similarity in neuronal correlation structures rather than similarity in states representation patterns (as in RSA) allows us to examine flexible knowledge representations when a mapping between states in the two tasks does not exist. We present three validations of this method: in cells, we show it captures all expected properties of grid and place cells, even if we reduce the data resolution by averaging over the activity of group of cells. In simulation, we show that calculating subspace generalization using simulated voxels from simulated grid cells results in significant generalization effect under realistic conditions. In fMRI, we show it captures the expected correlations induced by the visual properties of a task in LOC.

Our main finding of subspace generalization in EC across hexagonal graphs with different sizes and stimuli significantly strengthens the suggestion that EC flexibly represents all 'spatial-like' tasks, such as discretized 2D hexagonal graphs. Recently, we presented a theoretical framework for this idea: a neural network trained to predict future states that, when trained on 2D graphs displayed known spatial EC representations (the Tolman Eichenbaum Machine (TEM); *Whittington et al., 2020*). However, 'spatial-like' structures are not the only prevalent structures in natural tasks. The relations between task states often follow other structural forms (such as periodicities, hierarchies, or community structures), inference of which can aid behavior (*Mark et al., 2020*). Representations of non-Euclidean task structures have been found in EC (*Garvert et al., 2017*; *Baram et al., 2021*), and these generalize over different reinforcement learning tasks that are exactly the same except for their sensory properties (*Baram et al., 2021*). Indeed, when TEM was trained on non-Euclidean structures like hierarchical trees, it learned representations that were generalizable to novel environments with the same structure (*Whittington et al., 2020*). Further, we have previously shown that representing each family of graphs of the same structural form with the relevant stable representation (i.e. basis set) allows flexible transfer of the graph structure and therefore inference of unobserved transitions (relations between task's states) (*Mark et al., 2020*). Together, these studies suggest that flexible representation of structural knowledge may be encoded in the EC.

Based on these, we hypothesized that EC representations will also generalize over non-'spatial-like' tasks (here, community-structure) of different sizes. However, we could not find conclusive evidence for such a representation: the relevant contrast ([ClCl + ClCs + CsCl + CsCs] − [ClHl + ClHs + CsHl + CsHs]) did not yield a statistically significant effect in EC (or elsewhere, in an exploratory analysis corrected across the whole brain). This is despite clear behavioral evidence that participants use the community structure of the graph to inform their behavior: participants have a strong tendency to choose to move to the connecting nodes (nodes that connect two different communities) over non-connecting nodes (*Mark et al., 2020*, and *Appendix 1—figure 4a*). Moreover, in the post-experiment debriefing, participants could verbally describe the community structure of the graphs (26 out of 28 participants). This was not true for the hexagonal graphs. Why, then, did we not detect any neural generalization signals for the community structure graphs? There are both technical and psychological differences between the community structure and the hexagonal graphs that might have contributed to the difference in the results between the two structures. First, we have chosen a particular nested structure in which communities are organized on a ring. Subspace generalization may not be suitable for the detection of community structure: for example, a useful generalizable representation of such structure is composed of a binary 'within-community nodes' vs 'connecting nodes' representation. If this is the representation used by the brain, it means all 'community-encoding' voxels are similarly active in response to all stimuli (as all three-image sequences contain at least two non-connecting node images), and only 'connecting nodes encoding' voxels change their activation during stimuli presentation. Therefore, there is very little variance to detect.

Though this manuscript has focused on EC, it is worth noting that there is evidence for structural representations in other brain areas. Perhaps the most prominent of these is mPFC, where structural representations have been found in many contexts (*Klein-Flügge et al., 2019*; *Baram et al., 2021*; *Klein-Flügge et al., 2022*). Indeed, the strongest grid-like signals in abstract 2D tasks are often found in mPFC (*Constantinescu et al., 2016*; *Bao et al., 2019*; *Park et al., 2020*; *Bongioanni et al., 2021*) and task structure representations have been suggested to reside in mOFC (*Wilson et al., 2014*; *Schuck et al., 2016*; *Xie and Padoa-Schioppa, 2016*). The difference and interaction between PFC and MTL representations is a very active topic of research. One such suggested dissociation that might be of relevance here is the preferential contribution of MTL and PFC to latent and explicit learning, respectively. A related way of discussing this dissociation is to think of mPFC signals as closer to the deliberate actions subjects are taking. Circumstantial evidence from previous studies in our lab (tentatively) suggests the existence of such dissociation also for structural representations: when participants learned a graph structure without any awareness of it, this structure was represented in MTL but not mPFC (*Garvert et al., 2017*). On the other hand, when participants had to navigate on a 2D abstract graph to locations they were able to articulate, we observed much stronger grid-like signals in mPFC than MTL (though a signal in EC was also observed, *Constantinescu et al., 2016*). In addition, Baram et al. found that while the abstract structure of a reinforcement learning task was represented in EC, the structure-informed learning signals that inform trial-by-trial behavior with generalizable information were found in mPFC. Taken together, these results suggest that here, it is reasonable to expect generalization signals of community structure graphs (of which participants were aware) in PFC, as well as the signals reported in EC for hexagonal graphs (of which participants were unaware). Indeed, when we tested for subspace generalization of community structure graphs in the same ROI in vmPFC where Baram et al. found generalizable learning signals, we obtained a significant result (though this is a weak effect, and we hence report it with caution in Appendix 1, *Appendix 1—figure 4*).

To summarize, we have extended the understanding of EC representations and showed that EC represents hexagonal graph structures of different sizes, similarly to grid cells representation of spatial environments. We did this by using an analysis method which we believe will prove useful for the study of generalizable representations in different neural recording modalities. More work is needed to verify whether this principle of EC representations extends to other, non-'spatial-like' structural forms.

# Methods

## Rodent cells analysis

Cells electrophysiology data were taken from *Chen et al., 2018*. In short, cells (place cells from CA1 and grid cells from dmEC) were recorded while the animals foraged in two different square arenas; one real arena and one VR arena, real arena is 60 × 60 and the VR arena is 60 × 60 or 90 × 90 cm. The VR system restrained head movements to horizontal rotations and included an air-suspended ball on which the mice could run and turn. A virtual environment reflecting the mouse's movements on the ball was projected on screens in all horizontal directions and on the floor. Mice were implanted with custom-made microdrives (Axona, UK), loaded with 17 mm platinum–iridium tetrodes, and providing buffer amplification. We analyzed grid cells data from three animals; two animals had only grid cells data and one animal had both place cells and grid cells data. We analyzed place cells data from three more animals that had only place cells data (mouse 1 had 14 grid cells, mouse 2 and 3 had 21 grid cells, mouse 1, 4, and 5 had 25 place cells). This experimental design results in two different firing rate maps, one for each arena. After preprocessing (calculate the firing rate map using on 64 × 64 bins matrix and smoothing of the firing rate maps with 5 bins boxcar), we calculated the 'subspace generalization' score, as follows:

a. Calculate the neuron × neuron correlation matrix from the first firing rate map (one of the environments) and its PCs.
b. Project the firing rate maps from this environment and the other environment on these PCs.
c. Calculate the cumulative variance explained as a function of PCs (that are organized according to their corresponding eigenvalues).
d. Calculate the AUC.

## Permutation test 1 (within cell type)

Our hypothesis is that the neuron × neuron correlation structure is preserved while the animals forage in the two different arenas, that is that the active cells' assemblies remain the same. Therefore, the null hypothesis is that the cells' assemblies are random and did not remain the same while animals forage in the two arenas. We therefore calculated the PCs using the firing rate map while the animal foraged in one environment and permuted the cells' identity of the firing rate maps corresponding to the second environment. We then calculated the difference between the 'subspace generalization' score within and across environments. This creates our null distribution, which we compare to the subspace generalization score of the non-permuted data.

## Permutation test 2 (between cell types)

Our hypothesis is that grid cells generalize better than place cells, that is that the difference between the AUC of within arena projection to across arenas projection is smaller in grid cells compared to place cells. To this end, we created AUC-differences distribution using place cells activity as our null distribution; we sampled place cells from each animal, such that the number of grid cells and place cells was equal (mouse 1 had 14 grid cells, mouse 2 and 3 had 21 grid cells, and mouse 1, 4, and 5 had 25 place cells). Then, for each sample, we calculated the difference in AUC (same arena − different arenas), as before. We calculated the distribution of these AUC-difference values from all three animals. We then checked whether the AUC differences in grid cells, for all three animals, are significantly smaller than those predicted by the sampled place cells distribution (*Appendix 1—figure 1*).

## Reducing the resolution of the electrophysiological data

We first normalized all firing rate maps. Then, for each animal, we randomly sampled (with repeats) seven cells into two groups and averaged the cells' activity within each group, separately for each environment. We then concatenated the resulted size-2 vectors from all animals into one vector and used this vector as above to calculate the AUC differences between within and across environments. The number of bootstraps was 400; therefore, we had 800 repetitions to calculate the distribution (for each sample we project on both environments, therefore getting two AUC – difference values). The plots in *Figure 1d* were smoothed with a smoothing window of 9, the number of bins to calculate the distribution was 50.

## Simulating pseudo-voxels

Grid cells are simulated as a thresholded sum of three 2D cosines (*Burgess et al., 2007*). Each module is simulated by shifting the grid cells within a grid that spans the rhombus of the hexagonal grid, such that the average over all grid cells within a module is a constant across the box (note that due to numerical issues this is almost constant).

We simulated 13,456 cells per module (116*116 in the *x*–*y* plane, i.e. covering the grid's rhombus). The box is simulated with 50*50 resolution (the size of the 'box' is 10*10). We simulated four different modules that differ in their grid spacing and phases. Each environment was simulated by a different phase and shift of the grid fields such that the relationships between the cells remained the same across environments.

Voxels were simulated by averaging cells within a module. Each module was segregated into four groups of cells (therefore, there are 3364 cells within each voxel; see supplementary for different segregations). Each voxel is an average over the cells' firing rate map within the group. The averaging was done in two stages:

a. Sampling grid cells randomly – that is not related to their grid phase.
b. The remaining cells were segregated into four groups according to their phase.

The above process was repeated for different fractions of random/(according to phase) ratio (*ratio_random* = [0,1], 0: only segregated according to phase, 1: only segregated randomly). We further added spatial white noise to each voxel, noise std ranging from 0 to 0.1. When examining the effect of random sampling, the noise std was 0 or 0.1.

## fMRI experiment

### Participants

Sixty UCL students were originally recruited. As the training is long and hard, for each scan, we recruited two participants for the training sessions and chose the better performing of the two to be scanned. Overall, we scanned 34 participants and excluded 6 participants from the analysis because of severe movement or sleepiness in the scanner.

The study was approved by the University College London Research Ethics Committee (Project ID 11235/001). Participants gave written informed consent before the experiment.

## Behavioral training for fMRI training task

To ensure that participants understood the instructions, the first training day was performed in the lab while the other three training days were performed from the participant's home.

### Graphs

One hexagonal graph consisted of 36 nodes and the other 42 nodes as shown in *Figure 3B*. One community-structured graph consisted of five communities and the other six communities, with seven nodes each. Within a community, each node was connected to all other nodes except for the two connecting nodes that were not connected to each other but were each connected to a connecting node of a neighboring community (*Figure 3B*). Therefore, all nodes had a degree of six, similarly to the hexagonal graphs (except the nodes on the hexagonal graphs border, which had degree less than six). Our community structure graph had a hierarchical structure, wherein communities were organized on a ring.

### Training procedures

In each of the training days, participants learned two graphs with the same underlying structure but different stimuli. During the first 2 days, participants learned the hexagonal graphs, while during the third and fourth days, participants learned the community-structured graphs. We chose to first teach the hexagonal graphs structure for all participants and not randomize the order because learning community structure graph changes participants' learning strategy (*Mark et al., 2020*). During the fifth day, before the fMRI scan, participants were reminded of all four graphs, with two repetitions of each hexagonal graph and one repetition of each community-structured graph. Stimuli were selected randomly, for each participant, from a bank of stimuli (each pair of

graphs, one hexagonal and one of a community-structured graph, shared the same bank). Each graph was learned during four blocks (*Figure 3B*; four blocks for graph 1 followed by four blocks for graph 2 in each training day). Participants could take short resting breaks during the blocks. They were instructed to take a longer resting break after completing the four blocks of the first graph of each learning day.

## Block structure

Each block during training was made of the following tasks: (1) Learning phase, (2) Extending pictures sequences, (3) Can it be in the middle, (4) Navigation, and (5) Distance estimation (see *Figure 3*). Next, we elaborate on the various components of each block.

### Learning phase (Figure 3A)

Participants learned associations between graph nodes by observing a sequence of pairs of pictures which were sampled from a random walk on the graph (successive pairs of pictures shared a common picture). Participants were instructed to 'say something in their head' in order to remember the associations. Hexagonal graphs included 120 steps of the random walk per block and community-structured graphs included 180 steps per block (we introduced more pictures in the community graph condition as random walks on such graphs result in high sampling of transitions within a certain community and low sampling of transitions between communities).

### Extending pictures sequences (Figure 3D)

Given a target picture, which of two sequences of three pictures can be extended by that picture (a sequence can be extended by a picture only if it is a neighbor of the last picture in the sequence, the correct answer can be sequence 1/sequence 2/both sequences): 16 questions per block (note that a picture could not appear twice in the same sequence, that is if the target picture is already in the sequence the correct answer was necessarily the other sequence).

### Can it be in the middle (Figure 3C)

Determine whether a picture can appear between two other pictures, the answer is yes if and only if the picture is a neighbor of the two other pictures. Sixteen questions per block.

### Navigation (Figure 3E)

The aim – navigate to a target picture (appears at the right of the screen). The task was explained as a card game. Participants are informed that they currently have the card of the picture that appears on the left of the screen. They were asked to choose between two pictures that are associated with their current picture. They could also skip and sample again two pictures that are associated with the current picture, if they thought their two current options did not get them closer to the target (skipping was counted as a step). In each step, participants were instructed to choose a picture that they thought had a smaller number of steps to the target picture (according to their memory). Following the choice, the chosen picture appeared on the left and two new pictures, that correspond to states that are neighbors of the chosen picture, appeared as new choices. After a participant selected a neighbor of the target picture, that target picture itself could appear as one of the new options for choice. The game terminated when either the target was reached or 200 steps were taken (without reaching the target). In the latter case, a message 'too many steps' was displayed. On the first block, for each step, the number of links from the current picture to the target picture was shown on the screen. Participants played three games (i.e. navigation until the target was reached or 200 steps passed) in each block, where the starting distance (number of links) between the starting picture to the target was 2, 3, and 4.

## Distance estimation

Which of two pictures has the smallest number of steps to a target picture: 45 questions per block (none of the 2 pictures was a direct neighbor on the graph, i.e. the minimal distance was 2 and no feedback was given).

## fMRI scanning task

The task consisted of four runs. Each run was divided into five blocks (one block for each graph and one more repetition for one of the hexagonal graphs; the repetition was not used in the analyses in this manuscript). On each block, participants observed pictures that belong to one of the graphs. A block started with 70 s in which participants observed, at their own pace, a random walk on the graph; two neighboring pictures appeared on the screen, and when participants pressed 'enter' a new picture appeared on the screen (similar to the training learning phase). The new picture appeared in the middle of the screen, and the old picture appeared on its left. Participants were instructed to infer which 'pictures set' they were currently observing. No information about the graph was given. This random walk phase was not used in any analyses in this manuscript.

Next, sequences of three pictures appeared on the screen, one after the other (note the first and second pictures did not disappear from the screen until after the third picture in the sequence was presented – all three pictures disappeared together, prior to the next trial, *Figure 4b*). To keep participants engaged, once in a while (5 out of 45 sequences) a fourth picture appeared and participants had to indicate whether this picture can appear next on the sequence ('catch trials', *Figure 4c*). Before starting the fMRI scan, participants were asked whether they found any differences between the picture sets during the first 2 days (when the hexagonal graphs were learned) and the last 2 days (when the community graphs were learned). Most participants (26 out of 28) could indicate that there were groups of pictures (i.e. communities) in the last 2 days, and that this was not the case during the first 2 days. At the end of each block in the scanner, participants answered whether or not there are groups in the current picture set (participants that were not aware of the groups were asked whether this set belongs to the first two training days or not). Participants were given a bonus for answering correctly, such that 100% correct results in a ten-pound bonus.

## fMRI data acquisition

fMRI data was acquired on a 3T Siemens Prisma scanner using a 32 channels head coil. Functional scans were collected using a T2*-weighted echo-planar imaging (EPI) sequence with a multi-band acceleration factor of 4 (TR = 1.450 s, TE = 35 ms, flip angle = 70°, voxel resolution of 2 × 2 × 2 mm). A field map with dual echo-time images (TE1 = 10 ms, TE2 = 12.46 ms, whole-brain coverage, voxel size 2 × 2 × 2 mm) was acquired to correct for geometric distortions due to susceptibility-induced field inhomogeneities. Structural scans were acquired using a T1-weighted MPRAGE sequence with 1 × 1 × 1 mm voxel resolution. We discarded the first six volumes to allow for scanner equilibration.

## Preprocessing

Preprocessing was performed using tools from the fMRI Expert Analysis Tool (FEAT, *Woolrich et al., 2001*; *Woolrich et al., 2004*), part of FMRIB's Software Library (FSL, *Smith et al., 2004*). Data from each of the four scanner runs was preprocessed separately. Each run was aligned to a reference image using the motion correction tool MCFLIRT. Brain extraction was performed using the automated brain extraction tool BET (*Smith, 2002*). All data were temporally high-pass filtered with a cut-off of 100 s. Registration of EPI images to high-resolution structural images and to standard (MNI) space was performed using FMRIB's Linear Registration Tool (FLIRT *Jenkinson et al., 2002*; *Jenkinson and Smith, 2001*) and Non-Linear Registration Tool (FNIRT), respectively. No spatial smoothing was performed during preprocessing (see below for different smoothing protocols for each analysis). Because of the notable breathing- and susceptibility-related artifacts in the EC, we cleaned the data with FMRIB's ICA tool, FIX (*Griffanti et al., 2014*; *Salimi-Khorshidi et al., 2014*).

## Univariate analysis

Due to incompatibility of FSL with the MATLAB RSA toolbox (*Nili et al., 2014*) used in subsequent analyses, we estimated all first-level GLMs and univariate group-level analyses using SPM12 (Wellcome Trust Centre for Neuroimaging, https://www.fil.ion.ucl.ac.uk/spm/).

For estimating subspace generalization, we constructed a GLM to estimate the activation as a result of each three images' sequence (a 'pile' of pictures). The GLM includes the following regressors: mean CSF regressor and six motion parameters as nuisance regressors, bias term modeling the mean activity in each fMRI run, a regressor for the 'start' message (as a delta function), a regressor for the self-paced random walk on each graph (a delta function for each new picture that appears on the

screen), a regressor for each pile in each graph (duration of a pile: 1.4 s), regressor for the catch trial onset (delta) and the pile that corresponds to the catch (pile duration). All regressors besides the six motion regressors and CSF regressor were convolved with the HRF. The GLM was calculated using non-normalized data.

## Multivariate analysis

### Quantifying subspace generalization

We calculated noise normalized GLM betas within each searchlight using the RSA toolbox. For each searchlight and each graph, we had a nVoxels (100) by nPiles (10) activation matrix ($B_{voxel \times pile}$) that describes the activation of a voxel as a result of a particular pile (three pictures' sequence). We exploited the (voxel x voxel) covariance matrix of this matrix to quantify the manifold alignment within each searchlight.

To account for fMRI auto-correlation, we used the leave-one-out approach. For each fMRI scanner run and graph, we calculated the mean activation matrix over the three other scanner runs ($\hat{B}^{\sim j}$). We then calculated the left PCs of that matrix ($U^j_{voxel \times voxel}$). To quantify the alignment, we projected the excluded scanner run graph activation matrix ($B^j$) of each graph on these PCs and calculated the accumulated variance explained as a function of PCs, normalized by the total variance of each graph within each run. Therefore, for each run and graph, we calculated:

$$P^{a,b} = U_a^{\sim j} \cdot B_b^j$$

$$M_k^{a,b} = \frac{\sum_{l=1}^{10} \left( P_{a,b}^{l,k} \right)^2}{S^j}$$

$$\Sigma^j = U^{j^T} B^{j^T} B^j U^j$$

where $P^{a,b}$ is the projection matrix of dimensions $voxel \times pile$ of graph '$b$' on the PCs of graph '$a$', $M_k^{a,b}$ is the normalized variance explained on the '$k$' direction, $S^j$ is the summation of the diagonal of $\Sigma^j$, the total variance as a result of the graph piles (three-image sequence). We then calculated the cumulative variance explained over all '$k$' PCs directions. As a summary statistic, we calculated the area under this curve. This gives us a 4 × 4 alignment matrix, for each run, such that each entry ($a$, $b$) in this matrix is a measure of the alignment of voxels patterns as a result of the two graphs a and b (*Figure 4d*). We then averaged over the four runs and calculated different contrasts over this matrix.

The above calculations were performed in subject space; we therefore normalized the searchlight results and then smoothed them with a kernel of 6 mm FWHM using FSL FLIRT and FNIRT before performing group level statistics.

For group level, we calculated the *t*-stat over participants of each contrast:

Visual contrast was [HlHl + ClCl + HsHs + CsCs] – [HlHs + HsHl + ClCs + CsCl], that is same exact sequences controlled by the same structure.

Structural contrast was [HlHl + HlHs + HsHl + HsHs] – [HlCl + HlCs + HsCl + HsCs], that is the difference between subspace generalization of hexagonal graphs data, when projected on PCs calculated from (cross-validated) hexagonal graphs (yellow elements in middle panel) vs community structure graphs (red elements).

## Multiple comparisons correction

Multiple comparison correction was performed using the permutation tests machinery (*Nichols and Holmes, 2002*) in PALM (*Winkler et al., 2014*): within the mask we used for multiple comparisons correction (details in main text), we first measured the TFCE statistic for the current contrast. We then repeated this procedure for each of the 10,000 random sign-flip iterations (each participant's contrast sign was randomly flipped and the statistic over participants was calculated). Using these values, we then created a null distribution of TFCE statistics by saving only the voxel with the highest TFCE in each iteration. Comparing the true TFCE to the resulting null distributions results in FWE-corrected TFCE p-values.

## Acknowledgements

TB is supported by a Wellcome Principal Research Fellowship (219525/Z/19/Z), a Wellcome Collaborator award (214314/Z/18/Z), for the purpose of Open Access, the author has applied a CC BY public copyright license to any Author Accepted Manuscript version arising from this submission, a JS McDonnell Foundation award (JSMF220020372), and by the Jean Francois and Marie-Laure de Clermont Tonerre Foundation. The Wellcome Centre for Integrative Neuroimaging and Wellcome Centre for Human Neuroimaging are each supported by core funding from the Wellcome Trust (203139/Z/16/Z, 203147/Z/16/Z). The Sainsbury-Wellcome centre is supported by core funding from the Wellcome Trust (219627/Z/19/Z) and the Gatsby Charitable Foundation (GAT3755). AH was supported by the European Molecular Biology Organization nonstipendiary Long-Term Fellowship (848–2017), Human Frontier Science Program (LT000444/2018), Israeli National Postdoctoral Award Program for Advancing Women in Science, and the European Union's Horizon 2020 research and innovation programme under the Marie Skłodowska-Curie Grant Agreement No. 789040.

## Additional information

### Competing interests

Timothy E Behrens: Editor in Chief, *eLife*. The other authors declare that no competing interests exist.

### Funding

| Funder | Grant reference number | Author |
| --- | --- | --- |
| Wellcome Trust | 10.35802/219525 | Timothy E Behrens |
| Wellcome Trust | 10.35802/214314 | Timothy E Behrens |
| James S. McDonnell Foundation | 10.37717/220020372 | Timothy E Behrens |
| European Molecular Biology Organization | 848-2017 | Avital Hahamy |
| International Human Frontier Science Program Organization | LT000444/2018 | Avital Hahamy |
| Israeli National Postdoctoral Award Program for Advancing Women in Science | | Avital Hahamy |
| European Commission | 10.3030/789040 | Avital Hahamy |
| Jean Francois and Marie-Laure de Clermont Tonerre Foundation | | Timothy E Behrens |

The funders had no role in study design, data collection, and interpretation, or the decision to submit the work for publication. For the purpose of Open Access, the authors have applied a CC BY public copyright license to any Author Accepted Manuscript version arising from this submission.

### Author contributions

Shirley Mark, Conceptualization, Resources, Data curation, Software, Formal analysis, Validation, Investigation, Visualization, Methodology, Writing – original draft; Philipp Schwartenbeck, Data curation, Methodology; Avital Hahamy, Veronika Samborska, Methodology; Alon Boaz Baram, Visualization, Methodology, Writing – review and editing; Timothy E Behrens, Conceptualization, Formal analysis, Supervision, Funding acquisition, Methodology, Writing – review and editing

### Author ORCIDs

Shirley Mark ⓘ https://orcid.org/0000-0002-2160-1812
Philipp Schwartenbeck ⓘ https://orcid.org/0000-0001-8943-9965

Avital Hahamy  https://orcid.org/0000-0001-5862-851X
Alon Boaz Baram  https://orcid.org/0000-0002-6022-137X
Timothy E Behrens  https://orcid.org/0000-0003-0048-1177

### Ethics

Informed consent and consent to publish was obtained. All subjects gave written informed consent and the study was approved by the UCL ethics committee.

Reviewer #1 (Public review): https://doi.org/10.7554/eLife.101134.3.sa1
Reviewer #2 (Public review): https://doi.org/10.7554/eLife.101134.3.sa2
Reviewer #3 (Public review): https://doi.org/10.7554/eLife.101134.3.sa3
Author response https://doi.org/10.7554/eLife.101134.3.sa4

## Additional files

### Supplementary files

MDAR checklist

### Data availability

fMRI statistical maps have been deposited at: https://neurovault.org/collections/22681/. Anonymous behavioral data, code for the analysis and simulation have been deposited at: https://github.com/ShirleyMgit/subspace_generalization_paper_code (copy archived at *Mark, 2026*). The current guidelines of the Oxford Centre for Integrative Neuroimaging state that raw MRI data can not be considered fully anonymous, and hence at this moment can not be shared under GDPR. In addition, as the data for this study was acquired more than 5 years ago, and the consent forms signed by participants did not include the most up to date text regarding data sharing. This means that at the current moment we can only share data that is in normalised (MNI152) space, without the transformation matrices back to subject-spaces, and single subject human fMRI data cannot be uploaded. We are currently actively seeking guidance that will enable us to share the raw data. However, at this point we can not be certain we will be able to do so. An interested researcher should be in touch with Dr Alon Baram ( alon.baram@ndcn.ox.ac.uk) who will be able to give them the most up to date information about the status of the data sharing.

The following dataset was generated:

| Author(s) | Year | Dataset title | Dataset URL | Database and Identifier |
|---|---|---|---|---|
| Mark S | 2026 | Flexible neural representations of abstract structural knowledge in the human entorhinal cortex | https://neurovault.org/collections/22681/ | NeuroVault, 22681 |

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

## Appendix 1

## Grid and place cells analysis
Grid cells of all three animals generalize over the two different environments

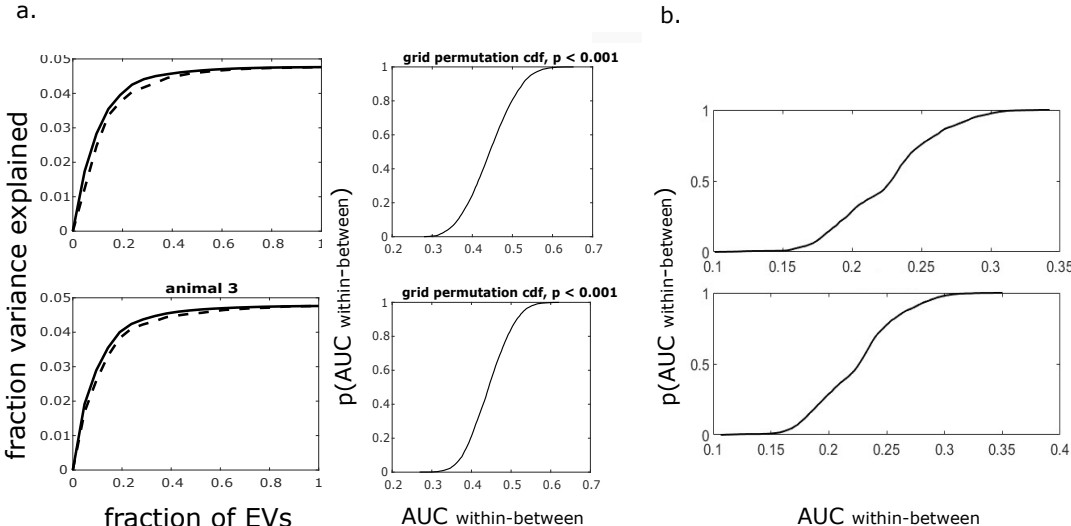

**Appendix 1—figure 1.** Subspace generalization: grid cells generalize over different environments. (**a**) Left: grid cells subspace generalization graph for two more mice (averaged over projection on environment 1 and 2). The lower and upper plots share the same legend. Right: permutation distributions of grid cells (number of permutations = 5000). We permuted the cells in the activity matrix (cells × bins) while the animal forages in one environment, then projected on the EVs from the activity matrix while the animal forages in the other environment and calculated the AUC (and vice versa). We then calculated the difference in AUC between the within-environment AUC and the AUC resulting from the permutation and calculated the cdf. Using these distributions, we can conclude that the difference in AUC of within and between environments is smaller than expected by random projections. p < 0.001 for both animals. (**b**) The place cells' sampling distribution of the difference in AUC of within and between arenas (number of bootstraps = 1000, for more details see methods, permutation test 2). We treat these distributions as our NULL distributions to answer the question whether grid cells' AUC within-between is smaller than place cells' AUC within-between that is do grid cells generalize significantly better than place cells. Upper plot: 14 cells are being sampled, lower plot: 21 cells are being sampled (to match the number of grid cells that were recorded within each animal), here again the calculation repeated twice, each time we project on one of the environments and then averaged the results.

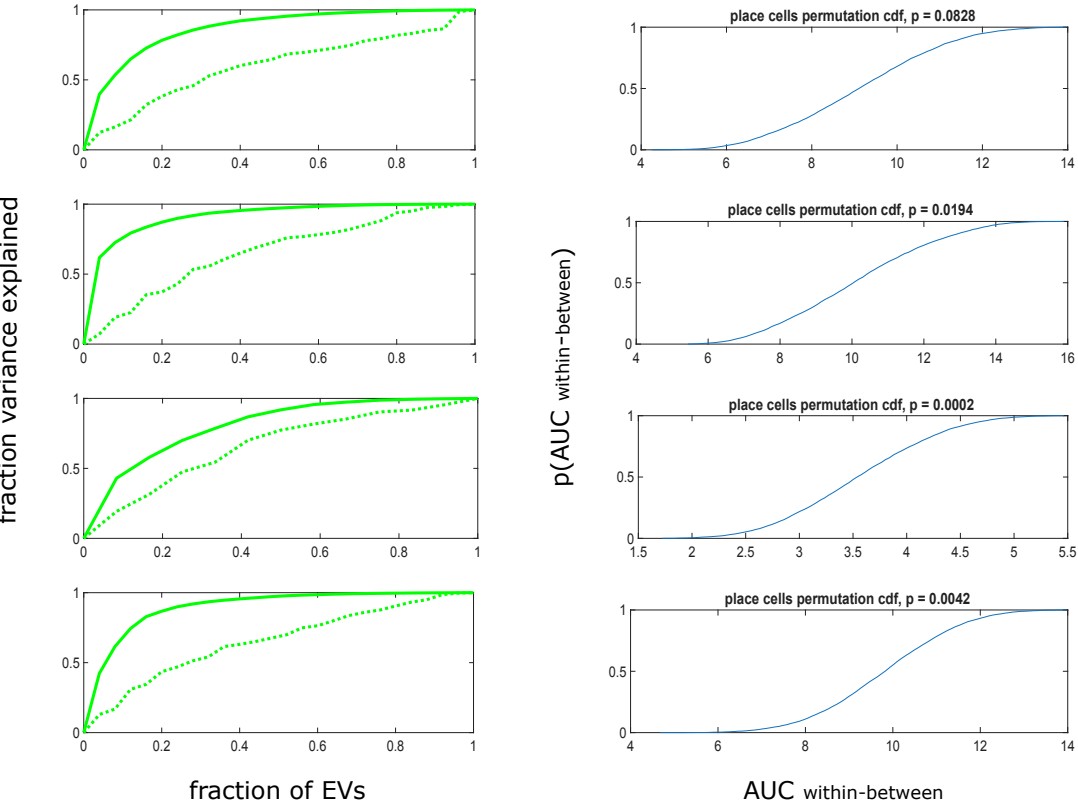

**Appendix 1—figure 2.** Subspace generalization of place cells. Left: subspace generalization plots for the different mice. Solid: within environment; dotted: across environments. Right: the corresponding permutation distribution within the corresponding mouse place cells. Cells were permuted before projection (as before). p-values correspond to the significance of the difference in the AUC for within and across environments under this perturbation distribution.

In all animals, the difference in AUC is smaller than expected by chance, though the effect is smaller than for grid cells. This contradicts the cartoon picture of orthogonal hippocampal representations between environments, but is consistent with models of hippocampal remapping (TEM, *Whittington et al., 2020*) and a number of recent empirical findings arguing that hippocampal remapping is non-random (*Liu et al., 2021*; *Samborska et al., 2022*; *Tanni et al., 2022*).

## From cells to voxels

We do not know a priori how the cells within a module are distributed into voxels. In the main manuscript, we segregate each module into four voxels. In *Appendix 1—figure 3a*, we show that we get similar results if we segregate the module into two voxels. Here, the cells are grouped into voxels according to the rhombus' diagonal. We next examine how the segregation of each module into a different number of voxels influences subspace-generalization score. Here, each voxel is composed of the sum over the cells' activity and noise. The signal-to-noise ratio (SNR) decreases as the number of voxels increases (*Appendix 1—figure 3b*) because the number of grid cells within a voxel decreases. The SNR is much larger for voxels with cells that are sampled according to phase compared to randomly (*Appendix 1—figure 3b*). Therefore, random sampling of cells to voxels leads to subspace generalization score within chance level (*Appendix 1—figure 3b*). We assume here no correlation between the noise across voxels, which may be unrealistic. A full investigation of the noise effect is out of the scope of this paper.

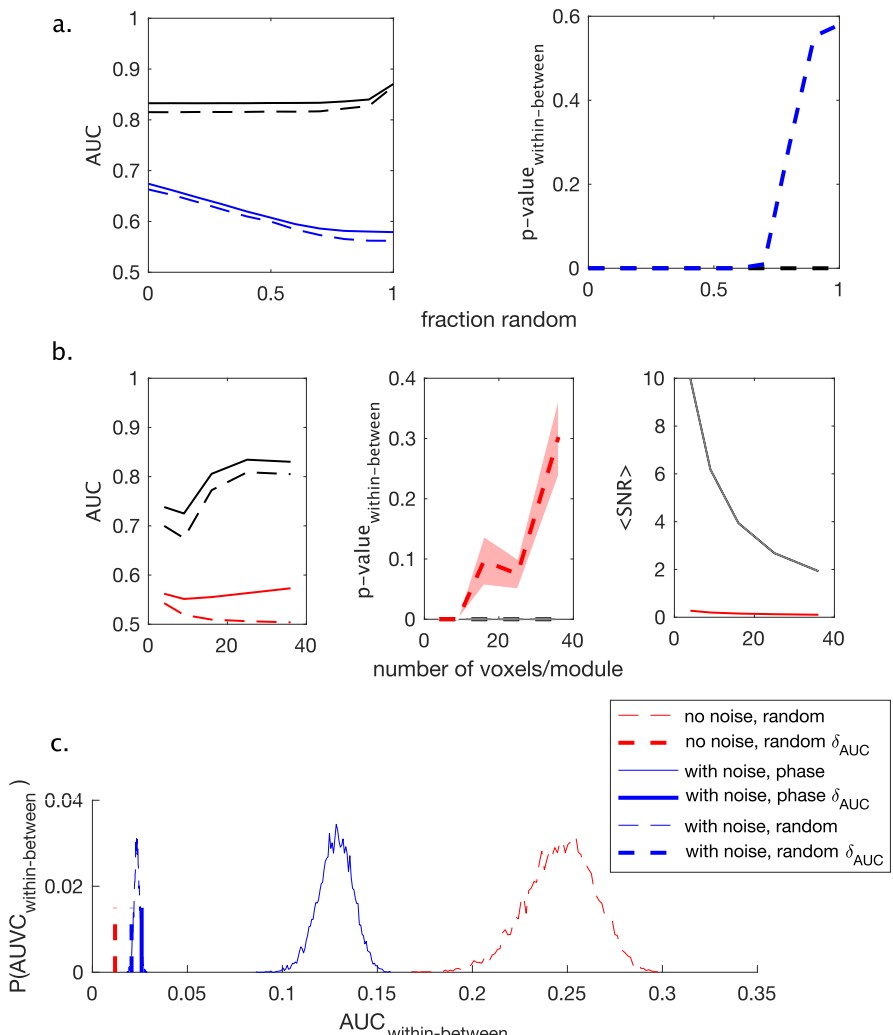

**Appendix 1—figure 3.** Dependency of subspace generalization on number of voxels per module. (**a**) Subspace generalization plot for voxels of the four modules, each module is segregated into two voxels as a function of the ratio of randomly sampled cells. Black – no noise, blue – with noise. Left: AUC (solid – within dash – across environments). Right: p-value of the effect according to the permutation distribution (see methods, shaded area – standard error of the mean). (**b**) Subspace generalization plot for voxels of 1095 of the 4 modules as a function of the number of voxels per module. Noise std = 50. Black – sampled according to phase, red – sampled randomly. (**c**) Left: AUC (solid – within dash – across environments). Middle: p-value of the effect according to the permutation distribution (see methods, shaded area – standard error of the mean). Right: signal-to-noise ratio (SNR) as a function of number of voxels per module, where the std (signal) is the averaged standard deviation over the voxels' activity map across the environment. Permutation distributions (the Null distribution). The shift of the distributions to the left following the introduction of noise/random sampling explains the increase in p-values even though the difference in AUC is still small.

## Subspace generalization vs correlating the neuron-by-neuron correlation matrices

In the following section, we compare subspace generalization as defined in our manuscript to a related and somewhat simpler method for looking at how neurons/voxels covary across tasks: correlating across tasks the (upper triangle of the) neuron-by-neuron correlation matrices. We show that subspace generalization emphasizes the low-dimensional characteristic of the representation and under reasonable conditions should be more robust to spatial noise correlations. Finally, we discuss the reasons for preferring to use the covariance matrix rather than the correlation matrix in the calculations of either method.

We can write the correlation between the covariance matrices, $C\alpha$, $C\beta$, as:

$$\rho = \frac{\text{tr}(C_\alpha C_\beta)}{\sqrt{\text{tr}(C_\alpha)\,\text{tr}(C_\beta)}} = \frac{\sum_q^N U_\alpha^q \left(C_\alpha C_\beta\right) U_\alpha^{q\,T}}{\sqrt{\text{tr}(C_\alpha)\,\text{tr}(C_\beta)}} = \frac{\sum_q^N \lambda_\alpha^q \left(U_\alpha^q C_\beta U_\alpha^{q\,T}\right)}{\sqrt{\text{tr}(C_\alpha)\,\text{tr}(C_\beta)}}$$

Because covariance matrices are symmetric, we can write the correlation between the upper triangular elements of these matrices as:

$$\rho = \frac{0.5}{\sqrt{\text{tr}(C_\alpha)\text{tr}(C_\beta)}} \left[\text{tr}(C_\alpha C_\beta) - \sum_i C_{ii}^\alpha C_{ii}^\beta\right] = 0.5 \left[\frac{\sum_q^N \lambda_\alpha^q (U_\alpha^q C_\beta U_\alpha^{q\,T})}{\sqrt{\text{tr}(C_\alpha)\text{tr}(C_\beta)}} - \sum_i C_{ii}^\alpha C_{ii}^\beta\right]$$

while the AUC in our method is calculated as:

$$\text{AUC} = \frac{\sum_q^N (N - q + 1)(U_\alpha^q C_\beta U_\alpha^{q\,T})}{\text{tr}(C_\beta)}$$

We can see that the differences between the two measures are (1) the scaling of each component in the summation (the projection of the matrix on the corresponding eigenvector) and (2) the effect of the diagonal elements of the matrix.

Emphasis on low-dimensional representation: We can look at the numerator of the two measures and define $f_\alpha^q \equiv U_\alpha^q C_\beta U_\alpha^{q\,T}$ then the numerator becomes:

$$\rho_{\text{numerator}} = \sum_q^N \lambda_\alpha^q f_\alpha^q \quad,$$

which can be thought of as the covariance between the eigenvector of one matrix and the projection (*f*), and

$$\text{AUC}_{\text{numerator}} = \sum_q^N \left(N - q + 1\right) f_\alpha^q$$

which can be thought of as a covariance of the projection *f* with a linear decreasing function. This emphasizes the requirement for decreased variance explained as a function of eigenvectors and therefore emphasizes the requirement for low-dimensional representation.

The diagonal elements of the covariance matrix give us the variance within each voxel/neuron along the different tasks' states; therefore, it is larger in areas that encode the states within the task and the signal is distributed across voxels.

Robustness to strong spatial noise: in the fMRI analysis, we look at the contrast between two conditions, one corresponds to our relevant conditions to compare (here, same structure) and the other corresponds to the control:

The difference in the correlation coefficient can be therefore written as:

$$\Delta\rho = \frac{1}{\sqrt{\text{tr}(C_\beta)}} \left(\frac{\sum_q^N \lambda_\alpha^q \left(U_\alpha^q C_\beta U_\alpha^{q,T}\right)}{\sqrt{\text{tr}(C_\alpha)}} - \frac{\sum_q^N \lambda_r^q \left(U_r^q C_\beta U_r^{q,T}\right)}{\sqrt{\text{tr}(C_r)}}\right)$$

while the difference in AUC can be written as:

$$\Delta\text{AUC} = \frac{\sum_q^N (N - q + 1) \cdot \left[\left(U_\alpha^q C_\beta U_\alpha^{q\,T}\right) - \left(U_r^q C_\beta U_r^{q\,T}\right)\right]}{\text{tr}(C_\beta)}$$

From the above, we can see that if there is a strong noise correlation, and it is orthogonal to the direction of the signal variance, it will be canceled out in our method (corresponding to the first eigenvector), while this is not necessarily the case in the difference between correlations (that requires not only the same ordinal position but the exact fraction of the corresponding eigenvalue and the sqrt of the trace). Therefore, in the case of strong spatial noise correlation, our method is

beneficial. Note that if there is strong spatial noise correlation in one condition and not the other, it will damage both methods, but because our study controls inputs that are not related to the structure, this seems unlikely. But please pay attention to the fact that our matrices $C\alpha$, $\beta$ are composed of covariance between the betas and not the pure voxels (see our response to the previous reviewer comment). Further research is needed in order to fully characterize which method is more beneficial and under which conditions.

Using correlation or covariance matrix to calculate the eigenvectors (which is the same question as whether we should standardize the data or not) is an interesting question that requires more thought and research. In our paper, we indeed use the covariance matrix that takes into account the scale of the data (voxel's beta/neuron firing rate). Using the covariance matrix indeed allows us to interpret our results as the direction of variance explained. But, it is a reasonable claim that the important measure to define a cell assembly (that may group into voxels) is the correlation between the neurons/voxels and not their covariation. Please note that in this case, the sum over the diagonal elements that is included in subspace generalization but ignored while correlating the upper triangular elements is just the dimension of the correlation matrix and therefore does not depend on the distribution of the data variance along the different dimension. Nevertheless, because here our variables are the betas from the first level GLM, it makes sense to give less weight, within a search light, on voxels with smaller betas which probably have very small SNR and that also means that their response to our stimuli is low.

### Inferring community structure
Learning a community structure: participants prefer to choose a connecting node even though it is the wrong answer

a.

b.

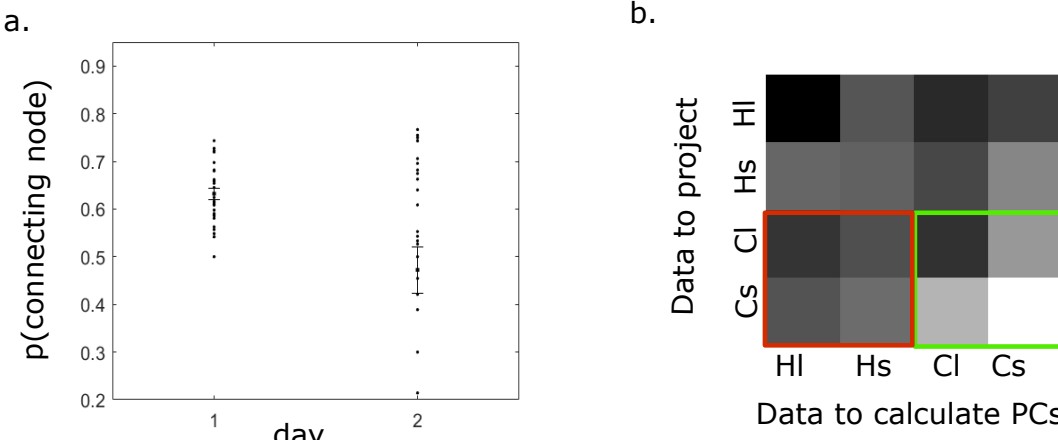

**Appendix 1—figure 4.** Learning a community structure. (**a**) During the first community structure training day, during the navigation task, participants preferred to choose connecting nodes even though it was the wrong answer. (**b**) Subspace generalization matrix in the PFC (ROI taken from ***Baram et al., 2021***, 125 voxels around 1171 the peak).

During the first day navigation task (see ***Figure 3F***), when participants had to choose between a picture that is a connecting node or a non-connecting node picture, they preferred to choose a connecting node picture even though it increases the number of steps to the target (***Appendix 1—figure 4a***, one-sided $t$-test against chance level (50%) $t(27) = 10.8$, $p < 0.001$). This result suggests that participants used the community structure graph to inform their behavior, i.e. the inference of the structure leads to a particular behavioral policy that fits that particular structure (see also ***Mark et al., 2020***). This was not true during the second day, when participants' knowledge of the graph improved; participants chose wrong connecting nodes during the first training day significantly more than during the second training day ($t(54) = 3.27$, $p < 0.01$).

## Flexible representation of structural knowledge in PFC

There is evidence of grid-like activity patterns in mPFC as well as EC while humans perform virtual navigation tasks (*Doeller et al., 2010*). The parallels between spatial and non-spatial task representations suggest that structural representation should be found in PFC as well, though it is still not clear what the conditions are that activate EC, PFC, or both. Indeed, a hexagonal activity pattern was also observed in PFC when humans navigated abstract 2D Euclidean space and when monkeys had to infer value-based decisions in non-spatial tasks (*Constantinescu et al., 2016*; *Bongioanni et al., 2021*). In our experiment, using the method that was described above, we could not find indications for PFC structural representation of hexagonal structure. One option is that PFC exploits structural representation when structural dependent decisions should be taken to gain a reward (*Baram et al., 2021*; *Samborska et al., 2022*).

In our experiment, there was no explicit reward. Nevertheless, navigating on a community-structured graph may result in internal reward; getting trapped inside a community is frustrating; therefore, escaping from a community is satisfying. Further, navigation on community-structured graphs requires the exploitation of a specific behavioral policy that is structure dependent (*Appendix 1—figure 3*, *Mark et al., 2020*). This raises the possibility that a flexible and abstract representation of community structure might exist in mPFC.

When humans had to make decisions in order to gain a reward while exploiting structural knowledge, previous study revealed a structure-dependent activation in mPFC (*Baram et al., 2021*). We therefore chose the ROI from Baram et al. to check for structural representation in our task. We have applied the same method as before on voxels in this ROI and checked for generalization of community structure knowledge. We indeed found that subspace generalization for community structure encoding in this ROI is significantly larger within community structure than between structure (i.e. the contrast: [ClCl + ClCs + CsCl + CsCs] − [ClHl + ClHs + CsHl + CsHs], coordinate: [−4, 44, −20], for this voxel $t(27) = 1.8$, $p < 0.05$, for the average of 125 voxels around the peak: $t(27) = 1.79$, $p < 0.05$, *Appendix 1—figure 4*), though we note that the effect is weak. Our results tentatively suggest that mPFC represents the knowledge of community-structured graphs abstractly and flexibly, but more experiments should be done to reinforce this conclusion.

## Sequences of three images in the graph

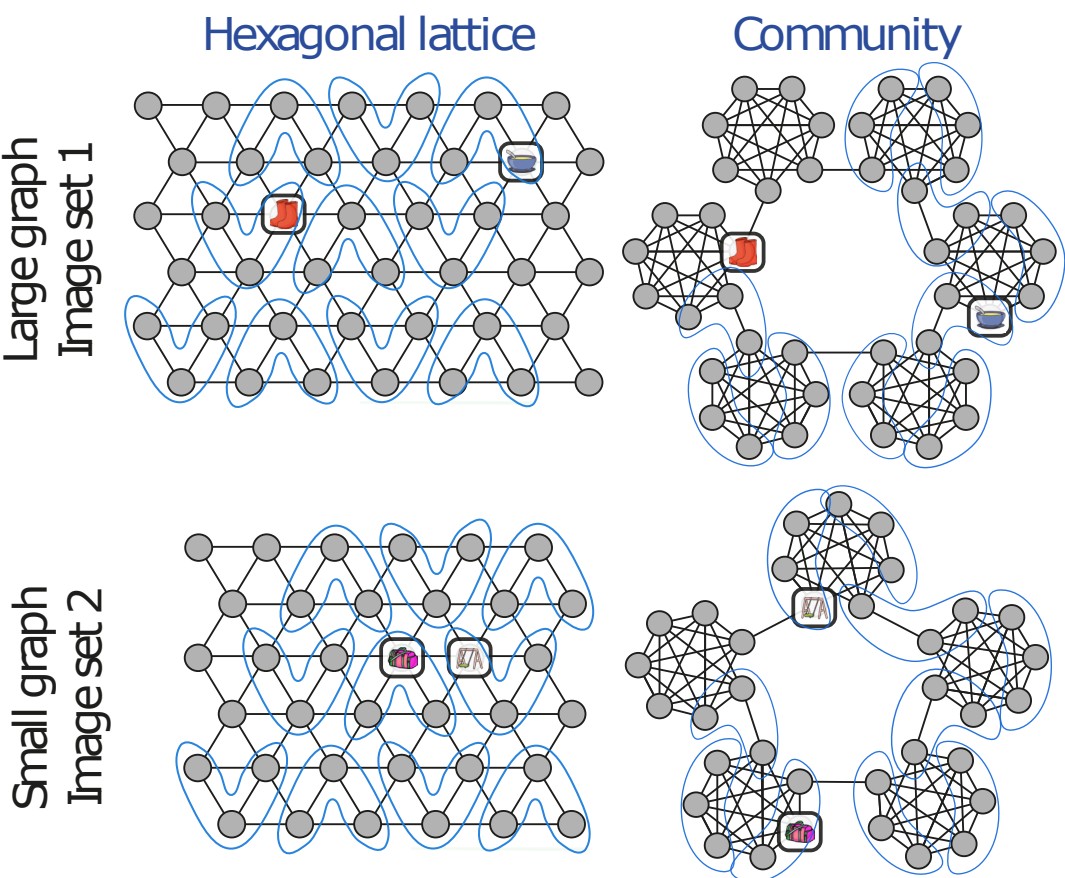

**Appendix 1—figure 5.** Partition of the graphs into three image sequences. Behavior during the fMRI session.

### Behavior during the fMRI session

To encourage attention to the sequences during the fMRI session, in 12.5% of trials the sequence was followed by a single image ('catch trial' in *Figure 4c*), and participants had to indicate whether it was associated with the last image in the sequence. Participants answered these questions significantly better than chance (*t*-test, p < 0.001, $t[27]_{hex}$ = 11.3, $t[27]_{cluster}$ = 10.6), for both types of structures, indicating that they: (1) remember the associations, (2) recognize the correct graph, and (3) maintain the correct representation during the block. On each block, we calculated the fraction of correct answers and averaged the fractions over all blocks within participant.

At the beginning of the fifth day, we asked participants whether they could describe how the images are associated, and whether they found any difference between the associations in the picture sets that were learnt during the first 2 days (i.e. the hexagonal graphs) and the sets that were learnt during the third and fourth day (the community graphs); 26 out of 28 participants mentioned verbally that the pictures in the sets of the last 2 days were segregated into groups (i.e. community structure). In the scanner, for these 26 participants, we asked at the end of each fMRI block whether the block's set of pictures contained groups. For the other two participants, we asked whether the set belonged to the first two training days. Below (Appendix—figure 6, right), we summarize participants' responses. Participants answered correctly which picture set they are currently playing with, significantly better than chance for both structures (*t*-test, p < 0.001 for both structures, $t[27]_{hex}$ = 3.8, $t[27]_{cluster}$ = 9.96).

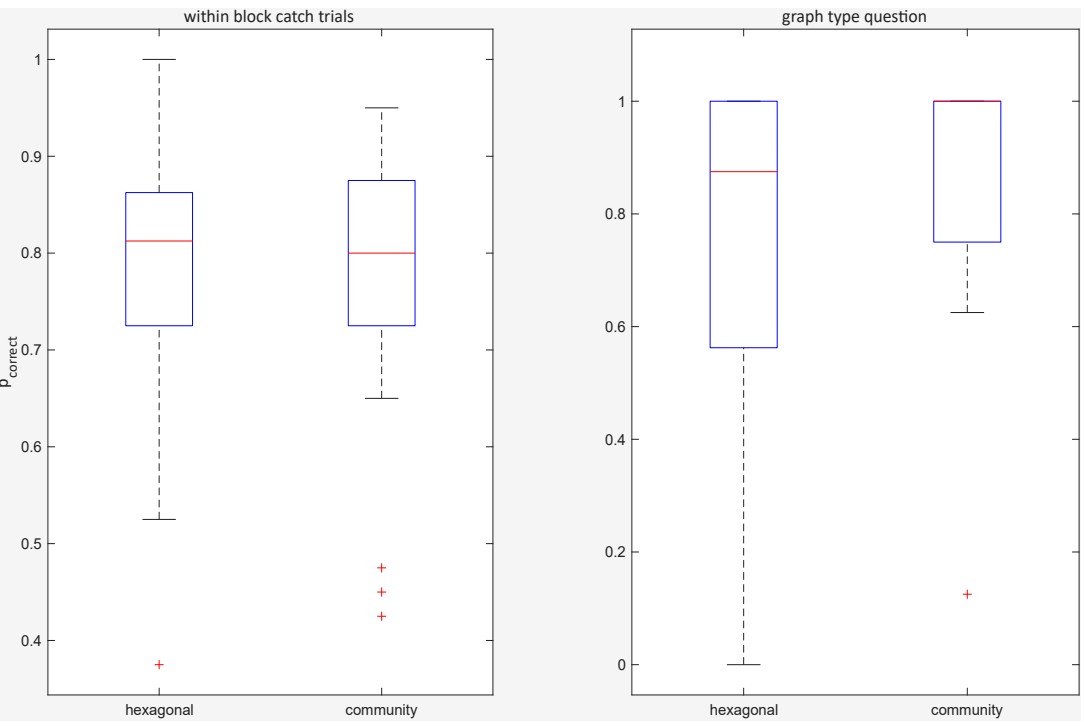

**Appendix 1—figure 6.** Behavior during the fMRI session. Left: Participants were able to detect whether the image in question can indeed follow the current three picture sequence significantly better than chance (*t*-test, p < 0.001, $t[27]_{hex}$ = 11.3, $t[27]_{cluster}$ = 10.6). Right: Fraction of correct answers to the recognition question at the end of each block (*t*-test, p < 0.001 for both structures, $t[27]_{hex}$ = 3.8, $t[27]_{cluster}$ = 9.96). Error bars denote SEMs.

