## [Editor Report · eLife Assessment]

Mark and colleagues developed and validated a **valuable** method for examining subspace generalization in fMRI data and applied it to understand whether the entorhinal cortex uses abstract representations that generalize across different environments with the same structure. The manuscript presents **convincing** evidence for the conclusion that abstract entorhinal representations of hexagonal associative structures generalize across different stimulus sets.

---

## [Referee Report · Reviewer #1 (Public review)]

Summary:

This study develops and validates a neural subspace similarity analysis for testing whether neural representations of graph structures generalize across graph size and stimulus sets. The authors show the method works in rat grid and place cell data, finding that grid but not place cells generalize across different environments, as expected. The authors then perform additional analyses and simulations to show that this method should also work on fMRI data. Finally, the authors test their method on fMRI responses from entorhinal cortex (EC) in a task that involves graphs that vary in size (and stimulus set) and statistical structure (hexagonal and community). They find neural representations of stimulus sets in lateral occipital complex (LOC) generalize across statistical structure and that EC activity generalizes across stimulus sets/graph size, but only for the hexagonal structures.

Strengths:

(1) The overall topic is very interesting and timely and the manuscript is well written.

(2) The method is clever and powerful. It could be important for future research testing whether neural representations are aligned across problems with different state manifestations.

(3) The findings provide new insights into generalizable neural representations of abstract task states in entorhinal cortex.

Weaknesses:

(1) There are two design confounds that are not sufficiently discussed.

(1.1) First, hexagonal and community structures are confounded by training order. All subjects learned the hexagonal graph always before the community graph. As such, any differences between the two graphs could be explained (in theory) by order effects (although this is unlikely). However, because community and hexagonal structures shared the same stimuli, it is possible that subjects had to find ways to represent the community structures separately from the hexagonal structures. This could potentially explain why there was no generalization across graph size for community structures.

(1.2) Second, subjects had more experience with the hexagonal and community structures before and during fMRI scanning. This is another possible reason why there was no generalization for the community structure.

(2) The authors include the results from a searchlight analysis to show specificity of the effects for EC. A more convincing way (in my opinion) to show specificity would be to test for (and report the results) of a double dissociation between the visual and structural contrast in two independently defined regions (e.g., anatomical ROIs of LOC and EC). This would substantiate the point that EC activity generalizes across structural similarity while sensory regions like LOC generalize across visual similarity.

---

## [Referee Report · Reviewer #2 (Public review)]

Summary:

Mark and colleagues test the hypothesis that entorhinal cortical representations may contain abstract structural information that facilitates generalization across structurally similar contexts. To do so, they use a method called "subspace generalization" designed to measure abstraction of representations across different settings. The authors validate the method using hippocampal place cells and entorhinal grid cells recorded in a spatial task, then show perform simulations that support that it might be useful in aggregated responses such as those measured with fMRI. Then the method is applied to an fMRI data that required participants to learn relationships between images in one of two structural motifs (hexagonal grids versus community structure). They show that the BOLD signal within an entorhinal ROI shows increased measures of subspace generalization across different tasks with the same hexagonal structure (as compared to tasks with different structures) but that there was not evidence for the complementary result (ie. increased generalization across tasks that share community structure, as compared to those with different structures). Taken together, this manuscript describes and validates a method for identifying fMRI representations that generalize across conditions and applies it to reveal that entorhinal representations that emerge across specific shared structural conditions.

Strengths:

I found this paper interesting both in terms of its methods and its motivating questions. The question asked is novel and the methods employed are new - and I believe this is the first time that they have been applied to fMRI data. I also found the iterative validation of the methodology to be interesting and important - showing persuasively that the method could detect a target representation - even in the face of random combination of tuning and with the addition of noise, both being major hurdles to investigating representations using fMRI.

Weaknesses:

The primary weakness of the paper in terms of empirical results is that the representations identified in EC had no clear relationship to behavior, raising questions about their functional importance.

The method developed is a clearly valuable tool that can serve as part of a larger battery of analysis techniques, but a small weakness on the methodological side is that for a given dataset, it might be hard to determine whether the method developed here would be better or worse than alternative methods.

---

## [Referee Report · Reviewer #3 (Public review)]

Summary:

The article explores the brain's ability to generalize information, with a specific focus on the entorhinal cortex (EC) and its role in learning and representing structural regularities that define relationships between entities in networks. The research provides empirical support for the longstanding theoretical and computational neuroscience hypothesis that the EC is crucial for structure generalization. It demonstrates that EC codes can generalize across non-spatial tasks that share common structural regularities, regardless of the similarity of sensory stimuli and network size.

Strengths:

At first glance, a potential limitation of this study appears to be its application of analytical methods originally developed for high-resolution animal electrophysiology (Samborska et al., 2022) to the relatively coarse and noisy signals of human fMRI. Rather than sidestepping this issue, however, the authors embrace it as a methodological challenge. They provide compelling empirical evidence and biologically grounded simulations to show that key generalization properties of entorhinal cortex representations can still be robustly detected. This not only validates their approach but also demonstrates how far non-invasive human neuroimaging can be pushed. The use of multiple independent datasets and carefully controlled permutation tests further underscores the reliability of their findings, making a strong case that structural generalization across diverse task environments can be meaningfully studied even in abstract, non-spatial domains that are otherwise difficult to investigate in animal models.

Weaknesses:

While this study provides compelling evidence for structural generalization in the entorhinal cortex (EC), several limitations remain that pave the way for promising future research. One issue is that the generalization effect was statistically robust in only one task condition, with weaker effects observed in the "community" condition. This raises the question of whether the null result genuinely reflects a lack of EC involvement, or whether it might be attributable to other factors such as task complexity, training order, or insufficient exposure possibilities that the authors acknowledge as open questions. Moreover, although the study leverages fMRI to examine EC representations in humans, it does not clarify which specific components of EC coding-such as grid cells versus other spatially tuned but non-grid codes-underlie the observed generalization. While electrophysiological data in animals have begun to address this, the human experiments do not disentangle the contributions of these different coding types. This leaves unresolved the important question of what makes EC representations uniquely suited for generalization, particularly given that similar effects were not observed in other regions known to contain grid cells, such as the medial prefrontal cortex (mPFC) or posterior cingulate cortex (PCC). These limitations point to important future directions for better characterizing the computational role of the EC and its distinctiveness within the broader network supporting learning and decision making based on cognitive maps.

---

## [Author Response]

The following is the authors’ response to the original reviews

**Public Reviews:**

**Reviewer #1 (Public review):**
Summary:This study develops and validates a neural subspace similarity analysis for testing whether neural representations of graph structures generalize across graph size and stimulus sets. The authors show the method works in rat grid and place cell data, finding that grid but not place cells generalize across different environments, as expected. The authors then perform additional analyses and simulations to show that this method should also work on fMRI data. Finally, the authors test their method on fMRI responses from the entorhinal cortex (EC) in a task that involves graphs that vary in size (and stimulus set) and statistical structure (hexagonal and community). They find neural representations of stimulus sets in lateral occipital complex (LOC) generalize across statistical structure and that EC activity generalizes across stimulus sets/graph size, but only for the hexagonal structures.Strengths:(1) The overall topic is very interesting and timely and the manuscript is well-written.(2) The method is clever and powerful. It could be important for future research testing whether neural representations are aligned across problems with different state manifestations.(3) The findings provide new insights into generalizable neural representations of abstract task states in the entorhinal cortex.

We thank the reviewer for their kind comments and clear summary of the paper and its strengths.

Weaknesses:(1) The manuscript would benefit from improving the figures. Moreover, the clarity could be strengthened by including conceptual/schematic figures illustrating the logic and steps of the method early in the paper. This could be combined with an illustration of the remapping properties of grid and place cells and how the method captures these properties.

We agree with the reviewer and have added a schematic figure of the method (figure 1a).

(2) Hexagonal and community structures appear to be confounded by training order. All subjects learned the hexagonal graph always before the community graph. As such, any differences between the two graphs could thus be explained (in theory) by order effects (although this is practically unlikely). However, given community and hexagonal structures shared the same stimuli, it is possible that subjects had to find ways to represent the community structures separately from the hexagonal structures. This could potentially explain why the authors did not find generalizations across graph sizes for community structures.

We thank the reviewer for their comments. We agree that the null result regarding the community structures does not mean that EC doesn’t generalise over these structures, and that the training order could in theory contribute to the lack of an effect. The decision to keep the asymmetry of the training order was deliberate: we chose this order based on our previous study (Mark et al. 2020), where we show that learning a community structure first changes the learning strategy of subsequent graphs. We could have perhaps overcome this by increasing the training periods, but (1) the training period is already very long; (2) there will still be asymmetry because the group that first learn community structure will struggle in learning the hexagonal graph more than vice versa, as shown in Mark et al. 2020.

We have added the following sentences on this decision to the Methods section:

“We chose to first teach hexagonal graphs for all participants and not randomize the order because of previous results showing that first learning community structure changes participants’ learning strategy (mark et al. 2020).”

(3) The authors include the results from a searchlight analysis to show the specificity of the effects of EC. A better way to show specificity would be to test for a double dissociation between the visual and structural contrast in two independently defined regions (e.g., anatomical ROIs of LOC and EC).

Thanks for this suggestion. We indeed tried to run the analysis in a whole-ROI approach, but this did not result in a significant effect in EC. Importantly, we disagree with the reviewer that this is a “better way to show specificity” than the searchlight approach. In our view, the two analyses differ with respect to the spatial extent of the representation they test for. The searchlight approach is testing for a highly localised representation on the scale of small spheres with only 100 voxels. The signal of such a localised representation is likely to be drowned in the noise in an analysis that includes thousands of voxels which mostly don’t show the effect - as would be the case in the whole-ROI approach.

(4) Subjects had more experience with the hexagonal and community structures before and during fMRI scanning. This is another confound, and possible reason why there was no generalization across stimulus sets for the community structure.

See our response to comment (2).

**Reviewer #2 (Public review):**
Summary:Mark and colleagues test the hypothesis that entorhinal cortical representations may contain abstract structural information that facilitates generalization across structurally similar contexts. To do so, they use a method called "subspace generalization" designed to measure abstraction of representations across different settings. The authors validate the method using hippocampal place cells and entorhinal grid cells recorded in a spatial task, then perform simulations that support that it might be useful in aggregated responses such as those measured with fMRI. Then the method is applied to fMRI data that required participants to learn relationships between images in one of two structural motifs (hexagonal grids versus community structure). They show that the BOLD signal within an entorhinal ROI shows increased measures of subspace generalization across different tasks with the same hexagonal structure (as compared to tasks with different structures) but that there was no evidence for the complementary result (ie. increased generalization across tasks that share community structure, as compared to those with different structures). Taken together, this manuscript describes and validates a method for identifying fMRI representations that generalize across conditions and applies it to reveal entorhinal representations that emerge across specific shared structural conditions.Strengths:I found this paper interesting both in terms of its methods and its motivating questions. The question asked is novel and the methods employed are new - and I believe this is the first time that they have been applied to fMRI data. I also found the iterative validation of the methodology to be interesting and important - showing persuasively that the method could detect a target representation - even in the face of a random combination of tuning and with the addition of noise, both being major hurdles to investigating representations using fMRI.

We thank the reviewer for their kind comments and the clear summary of our paper.

Weaknesses:In part because of the thorough validation procedures, the paper came across to me as a bit of a hybrid between a methods paper and an empirical one. However, I have some concerns, both on the methods development/validation side, and on the empirical application side, which I believe limit what one can take away from the studies performed.

We thank the reviewer for the comment. We agree that the paper comes across as a bit of a methods-empirical hybrid. We chose to do this because we believe (as the reviewer also points out) that there is value in both aspects of the paper.

Regarding the methods side, while I can appreciate that the authors show how the subspace generalization method "could" identify representations of theoretical interest, I felt like there was a noticeable lack of characterization of the specificity of the method. Based on the main equation in the results section of the paper, it seems like the primary measure used here would be sensitive to overall firing rates/voxel activations, variance within specific neurons/voxels, and overall levels of correlation among neurons/voxels. While I believe that reasonable pre-processing strategies could deal with the first two potential issues, the third seems a bit more problematic - as obligate correlations among neurons/voxels surely exist in the brain and persist across context boundaries that are not achieving any sort of generalization (for example neurons that receive common input, or voxels that share spatial noise). The comparative approach (ie. computing difference in the measure across different comparison conditions) helps to mitigate this concern to some degree - but not completely - since if one of the conditions pushes activity into strongly spatially correlated dimensions, as would be expected if univariate activations were responsive to the conditions, then you'd expect generalization (driven by shared univariate activation of many voxels) to be specific to that set of conditions.

We thank the reviewer for their comments. We would like to point out that we demean each voxel within all states/piles (3-pictures sequences) in a given graph/task (what the reviewer is calling “a condition”). Hence there is no shared univariate activation of many voxels in response to a graph going into the computation, and no sensitivity to the overall firing rate/voxel activation. Our calculation captures the variance across states conditions within a task (here a graph), over and above the univariate effect of graph activity. In addition, we spatially pre-whiten the data within each searchlight, meaning that noisy voxels with high noise variance will be downweighted and noise correlations between voxels are removed prior to applying our method.

A second issue in terms of the method is that there is no comparison to simpler available methods. For example, given the aims of the paper, and the introduction of the method, I would have expected the authors to take the Neuron-by-Neuron correlation matrices for two conditions of interest, and examine how similar they are to one another, for example by correlating their lower triangle elements. Presumably, this method would pick up on most of the same things - although it would notably avoid interpreting high overall correlations as "generalization" - and perhaps paint a clearer picture of exactly what aspects of correlation structure are shared. Would this method pick up on the same things shown here? Is there a reason to use one method over the other?

We thank the reviewer for this important and interesting point. We agree that calculating correlation between the upper triangular elements of the covariance or correlation matrices picks up similar, but not identical aspects of the data (see below the mathematical explanation that was added to the supplementary). When we repeated the searchlight analysis and calculated the correlation between the upper triangular entries of the Pearson correlation matrices we obtained an effect in the EC, though weaker than with our subspace generalization method (t=3.9, the effect did not survive multiple comparisons). Similar results were obtained with the correlation between the upper triangular elements of the covariance matrices(t=3.8, the effect did not survive multiple comparisons).

The difference between the two methods is twofold: (1) Our method is based on the covariance matrix and not the correlation matrix - i.e. a difference in normalisation. We realised that in the main text of the original paper we mistakenly wrote “correlation matrix” rather than “covariance matrix” (though our equations did correctly show the covariance matrix). We have corrected this mistake in the revised manuscript. (2) The weighting of the variance explained in the direction of each eigenvector is different between the methods, with some benefits of our method for identifying low-dimensional representations and for robustness to strong spatial correlations. We have added a section “Subspace Generalisation vs correlating the Neuron-by-Neuron correlation matrices” to the supplementary information with a mathematical explanation of these differences.

Regarding the fMRI empirical results, I have several concerns, some of which relate to concerns with the method itself described above. First, the spatial correlation patterns in fMRI data tend to be broad and will differ across conditions depending on variability in univariate responses (ie. if a condition contains some trials that evoke large univariate activations and others that evoke small univariate activations in the region). Are the eigenvectors that are shared across conditions capturing spatial patterns in voxel activations? Or, related to another concern with the method, are they capturing changing correlations across the entire set of voxels going into the analysis? As you might expect if the dynamic range of activations in the region is larger in one condition than the other?

This is a searchlight analysis, therefore it captures the activity patterns within nearby voxels. Indeed, as we show in our simulation, areas with high activity and therefore high signal to noise will have better signal in our method as well. Note that this is true of most measures.

My second concern is, beyond the specificity of the results, they provide only modest evidence for the key claims in the paper. The authors show a statistically significant result in the Entorhinal Cortex in one out of two conditions that they hypothesized they would see it. However, the effect is not particularly large. There is currently no examination of what the actual eigenvectors that transfer are doing/look like/are representing, nor how the degree of subspace generalization in EC may relate to individual differences in behavior, making it hard to assess the functional role of the relationship. So, at the end of the day, while the methods developed are interesting and potentially useful, I found the contributions to our understanding of EC representations to be somewhat limited.

We agree with this point, yet believe that the results still shed light on EC functionality. Unfortunately, we could not find correlation between behavioral measures and the fMRI effect.

**Reviewer #3 (Public review):**
Summary:The article explores the brain's ability to generalize information, with a specific focus on the entorhinal cortex (EC) and its role in learning and representing structural regularities that define relationships between entities in networks. The research provides empirical support for the longstanding theoretical and computational neuroscience hypothesis that the EC is crucial for structure generalization. It demonstrates that EC codes can generalize across non-spatial tasks that share common structural regularities, regardless of the similarity of sensory stimuli and network size.Strengths:(1) Empirical Support: The study provides strong empirical evidence for the theoretical and computational neuroscience argument about the EC's role in structure generalization.(2) Novel Approach: The research uses an innovative methodology and applies the same methods to three independent data sets, enhancing the robustness and reliability of the findings.(3) Controlled Analysis: The results are robust against well-controlled data and/or permutations.(4) Generalizability: By integrating data from different sources, the study offers a comprehensive understanding of the EC's role, strengthening the overall evidence supporting structural generalization across different task environments.Weaknesses:A potential criticism might arise from the fact that the authors applied innovative methods originally used in animal electrophysiology data (Samborska et al., 2022) to noisy fMRI signals. While this is a valid point, it is noteworthy that the authors provide robust simulations suggesting that the generalization properties in EC representations can be detected even in low-resolution, noisy data under biologically plausible assumptions. I believe this is actually an advantage of the study, as it demonstrates the extent to which we can explore how the brain generalizes structural knowledge across different task environments in humans using fMRI. This is crucial for addressing the brain's ability in non-spatial abstract tasks, which are difficult to test in animal models.While focusing on the role of the EC, this study does not extensively address whether other brain areas known to contain grid cells, such as the mPFC and PCC, also exhibit generalizable properties. Additionally, it remains unclear whether the EC encodes unique properties that differ from those of other systems. As the authors noted in the discussion, I believe this is an important question for future research.

We thank the reviewer for their comments. We agree with the reviewer that this is a very interesting question. We tried to look for effects in the mPFC, but we did not obtain results that were strong enough to report in the main manuscript, but we do report a small effect in the supplementary.

**Recommendations for the authors:**

**Reviewer #1 (Recommendations for the authors):**
(1) I wonder how important the PCA on B1(voxel-by-state matrix from environment 1) and the computation of the AUC (from the projection on B2 [voxel-by-state matrix from environment 1]) is for the analysis to work. Would you not get the same result if you correlated the voxel-by-voxel correlation matrix based on B1 (C1) with the voxel-by-voxel correlation matrix based on B2 (C2)? I understand that you would not have the subspace-by-subspace resolution that comes from the individual eigenvectors, but would the AUC not strongly correlate with the correlation between C1 and C2?

We agree with the reviewer comments - see our response to reviewer 2 second issue above.

(2) There is a subtle difference between how the method is described for the neural recording and fMRI data. Line 695 states that principal components of the neuron x neuron intercorrelation matrix are computed, whereas line 888 implies that principal components of the data matrix B are computed. Of note, B is a voxel x pile rather than a pile x voxel matrix. Wouldn't this result in U being pile x pile rather than voxel x voxel?

The PCs are calculated on the neuron x neuron (or voxel x voxel) covariance matrix of the activation matrix. We’ve added the following clarification to the relevant part of the Methods:

“We calculated noise normalized GLM betas within each searchlight using the RSA toolbox. For each searchlight and each graph, we had a nVoxels (100) by nPiles (10) activation matrix (B) that describes the activation of a voxel as a result of a particular pile (three pictures’ sequence). We exploited the (voxel x voxel) covariance matrix of this matrix to quantify the manifold alignment within each searchlight.”

(3) It would be very helpful to the field if the authors would make the code and data publicly available. Please consider depositing the code for data analysis and simulations, as well as the preprocessed/extracted data for the key results (rat data/fMRI ROI data) into a publicly accessible repository.

The code is publicly available in git (https://github.com/ShirleyMgit/subspace_generalization_paper_code/tree/main).

(4) Line 219: "Kolmogorov Simonov test" should be "Kolmogorov Smirnov test".

thanks!

(5) Please put plots in Figure 3F on the same y-axis.(6) Were large and small graphs of a given statistical structure learned on the same days, and if so, sequentially or simultaneously? This could be clarified.

The graphs are learned on the same day. We clarified this in the Methods section.

**Reviewer #2 (Recommendations for the authors):**
Perhaps the advantage of the method described here is that you could narrow things down to the specific eigenvector that is doing the heavy lifting in terms of generalization... and then you could look at that eigenvector to see what aspect of the covariance structure persists across conditions of interest. For example, is it just the highest eigenvalue eigenvector that is likely picking up on correlations across the entire neural population? Or is there something more specific going on? One could start to get at this by looking at Figures 1A and 1C - for example, the primary difference for within/between condition generalization in 1C seems to emerge with the first component, and not much changes after that, perhaps suggesting that in this case, the analysis may be picking up on something like the overall level of correlations within different conditions, rather than a more specific pattern of correlations.

The nature of the analysis means the eigenvectors are organized by their contribution to the variance, therefore the first eigenvector is responsible for more variance than the other, we did not check rigorously whether the variance is then splitted equally by the remaining eigenvectors but it does not seems to be the case.

Why is variance explained above zero for fraction EVs = 0 for figure 1C (but not 1A) ? Is there some plotting convention that I'm missing here?

There was a small bug in this plot and it was corrected - thank you very much!

The authors say:"Interestingly, the difference in AUCs was also 190 significantly smaller than chance for place cells (Figure 1a, compare dotted and solid green 191 lines, p<0.05 using permutation tests, see statistics and further examples in supplementary 192 material Figure S2), consistent with recent models predicting hippocampal remapping that is 193 not fully random (Whittington et al. 2020)."But my read of the Whittington model is that it would predict slight positive relationships here, rather than the observed negative ones, akin to what one would expect if hippocampal neurons reflect a nonlinear summation of a broad swath of entorhinal inputs.

Smaller differences than chance imply that the remapping of place cells is not completely random.

Figure 2:I didn't see any description of where noise amplitude values came from - or any justification at all in that section. Clearly, the amount of noise will be critical for putting limits on what can and cannot be detected with the method - I think this is worthy of characterization and explanation. In general, more information about the simulations is necessary to understand what was done in the pseudovoxel simulations. I get the gist of what was done, but these methods should clear enough that someone could repeat them, and they currently are not.

Thanks, we added noise amplitude to the figure legend and Methods.

What does flexible mean in the title? The analysis only worked for the hexagonal grid - doesn't that suggest that whatever representations are uncovered here are not flexible in the sense of being able to encode different things?

Flexible here means, flexible over stimulus’ characteristics that are not related to the structural form such as stimuli, the size of the graph etc.

**Reviewer #3 (Recommendations for the authors):**
I have noticed that the authors have updated the previous preprint version to include extensive simulations. I believe this addition helps address potential criticisms regarding the signal-to-noise ratio. If the authors could share the code for the fMRI data and the simulations in an open repository, it would enhance the study's impact by reaching a broader readership across various research fields. Except for that, I have nothing to ask for revision.

Thanks, the code will be publicly available: (https://github.com/ShirleyMgit/subspace_generalization_paper_code/tree/main).